# Two-way Deconfounder for Off-policy Evaluation in Causal Reinforcement Learning

**Shuguang Yu**[*]
School of Statistics and Management
Shanghai University of Finance and Economics
Shanghai, China

**Shuxing Fang**[*]
Department of Applied Mathematics
The Hong Kong Polytechnic University
Hong Kong, China

**Ruixin Peng**
School of Statistics and Management
Shanghai University of Finance and Economics
Shanghai, China

**Zhengling Qi**
Department of Decision Sciences
George Washington University
Washington D.C., USA

**Fan Zhou**[†]
School of Statistics and Management
Shanghai University of Finance and Economics
Shanghai, China

**Chengchun Shi**[†]
Department of Statistics
London School of Economics and Political Science
London, UK

## Abstract

This paper studies off-policy evaluation (OPE) in the presence of unmeasured confounders. Inspired by the two-way fixed effects regression model widely used in the panel data literature, we propose a two-way unmeasured confounding assumption to model the system dynamics in causal reinforcement learning and develop a two-way deconfounder algorithm that devises a neural tensor network to simultaneously learn both the unmeasured confounders and the system dynamics, based on which a model-based estimator can be constructed for consistent policy value estimation. We illustrate the effectiveness of the proposed estimator through theoretical results and numerical experiments.

## 1   Introduction

Before deploying any newly developed policy, it is important to assess its impact. In many high-stakes domains, it is risky or unethical to implement such policies directly for online evaluation. This challenge highlights the essential role of off-policy evaluation (OPE).

There is a vast body of literature on OPE. Most studies assume there are no unmeasured confounders (NUC), also known as unconfoundedness (see e.g., Thomas et al. 2015, Jiang and Li 2016, Thomas and Brunskill 2016, Farajtabar et al. 2018, Liu et al. 2018, Irpan et al. 2019, Schlegel et al. 2019, Tang et al. 2019, Xie et al. 2019, Dai et al. 2020, Chandak et al. 2021, Hao et al. 2021, Liao et al. 2021b,

---

[*]Equal contribution.
[†]Corresponding authors: `zhoufan@mail.shufe.edu.cn`, `c.shi7@lse.ac.uk`.

38th Conference on Neural Information Processing Systems (NeurIPS 2024).

Shi et al. 2021, Chen and Qi 2022, Kallus and Uehara 2022, Liao et al. 2022, Shi et al. 2022b, Xie et al. 2023, Zhou et al. 2023a). However, the NUC assumption is restrictive and untestable from the data. It can be violated in various domains such as urgent care [Namkoong et al., 2020], autonomous driving [Nyholm and Smids, 2020], ride-sharing [Shi et al., 2022c], and bidding [Xu et al., 2023]. Applying standard OPE methods that rely on the NUC assumption in these settings would result in a biased policy value estimator [see e.g., Bennett and Kallus, 2023, Section 6.1].

Causal reinforcement learning studies offline policy optimization or OPE in the presence of unmeasured confounding. Many existing works can be divided into one of the following three groups:

1. ***Methods under memoryless unmeasured confounding***: The first type of methods relies on a "memoryless unmeasured confounding" assumption to guarantee that the observed data satisfies the Markov property in the presence of unmeasured confounders [Zhang and Bareinboim, 2016, Kallus and Zhou, 2020, Li et al., 2021, Liao et al., 2021a, Wang et al., 2021, Chen et al., 2022, Fu et al., 2022, Shi et al., 2022c, Yu et al., 2022, Bruns-Smith and Zhou, 2023, Xu et al., 2023]. Many of these works also require external proxy variables (e.g., mediators and instrumental variables) to handle latent confounders. In contrast, the method proposed in this article neither relies on the Markov assumption nor requires external proxies.

2. ***POMDP-type methods***: The second category employs a partially observable Markov decision process (POMDP) to model unmeasured confounders as latent states, drawing on ideas from the proximal causal inference literature [Tchetgen et al., 2020] to address unmeasured confounding [Tennenholtz et al., 2020, Nair and Jiang, 2021, Miao et al., 2022, Shi et al., 2022a, Wang et al., 2022, Bennett and Kallus, 2023, Hong et al., 2023, Lu et al., 2023]. However, these works require restrict mathematical assumptions that are hard to verify in practice [Lu et al., 2018].

3. ***Deconfounding-type methods***: The final category originates from deconfounding methods in causal inference, which leverage the inherent structure within observed data, such as multiple treatment dependencies, network structures, and exposure models, to address unmeasured confounding [Louizos et al., 2017, Tran and Blei, 2017, Wang et al., 2018, Veitch et al., 2019, Wang and Blei, 2019, Zhang et al., 2019, Bica et al., 2020, Veitch et al., 2020, Shah et al., 2022, McFowland III and Shalizi, 2023, Shuai et al., 2023]. Notably, Wang and Blei [2019] directly estimate latent confounders for causal inference. However, their algorithm requires the unmeasured confounders to be a deterministic function of the actions [Ogburn et al., 2020, Wang and Blei, 2019], which has been criticized as being unreasonable [D'Amour, 2019, Ogburn et al., 2019]; refer to Appendix A.1. There have also been several extensions of deconfounding methods to reinforcement learning (RL) [Lu et al., 2018, Hatt and Feuerriegel, 2021, Kausik et al., 2022, 2023]. However, these methods either impose a one-way unmeasured confounding assumption, which can be overly restrictive (see Section 2), or require the correct specification of the latent variable [Rissanen and Marttinen, 2021].

This paper aims to develop advanced deconfounding-type OPE methodologies, allowing for more flexible assumptions regarding latent variable modeling. Our proposal is inspired by the two-way fixed effects (2FE) model which is widely employed in applied economics Mundlak [1961], Baltagi and Baltagi [2008], Griliches [1979], Anderson and Hsiao [1982], Freyberger [2018], Callaway and Karami [2023] and causal inference with panel data Arkhangelsky and Imbens [2022], De Chaisemartin and d'Haultfoeuille [2020], Imai and Kim [2021], Sant'Anna and Zhao [2020], Athey and Imbens [2022]. More recently, Dwivedi et al. [2022] applied the 2FE model to counterfactual prediction in a contextual bandit setting and Bian et al. [2023] extended the 2FE model to RL. However, their investigations primarily consider an unconfounded setting. Additionally, the model proposed by Bian et al. [2023] imposes a restrictive additive assumption – requiring the latent factors to influence both the reward and transition functions in a purely additive manner. Furthermore, they rely on linear function approximation to estimate the policy value, which may fail to capture the inherently complex nonlinear dynamics. In contrast, we employ flexible neural networks to model the environment.

In this article, we propose a novel two-way unmeasured confounding assumption to effectively model latent confounders. This approach categorizes all unmeasured confounders into two distinct groups: those that are time-specific and those that are trajectory-specific. This assumption enhances the model's flexibility beyond one-way unmeasured confounding while ensuring that the total number of confounders remains much smaller than the sample size, making them estimable from the observed data. We further develop an original two-way deconfounder algorithm that constructs a neural tensor network to jointly learn the unmeasured confounders and the system dynamics. Based on the learned

model, we construct a model-based estimator to accurately estimate the policy value. Our proposed model for unmeasured confounders shares similarities with latent factor models used to capture two-way interactions, such as item-customer interactions in recommendation systems [Hu et al., 2008, He et al., 2017], and relationships between different entities in multi-relational learning [Socher et al., 2013, Nickel et al., 2015, Nguyen et al., 2017, Wang et al., 2017, Ji et al., 2021].

To summarize, our contributions include: (1) the introduction of a novel two-way unmeasured confounding assumption; (2) the development of a new two-way deconfounder algorithm for model-based OPE under unmeasured confounding; (3) the demonstration of the effectiveness of our model and algorithm.

## 2 Two-way Unmeasured Confounding

In this section, we begin by presenting the data generating process under unmeasured confounding and outlining our objectives. We then introduce the proposed two-way unmeasured confounding assumption and compare it against alternative assumptions.

We consider an offline setting with pre-collected observational data $\mathcal{D}$, containing $N$ trajectories, each consisting of $T$ time points. We use $i$ to index the $i$-th trajectory and $t$ to index the $t$-th time point. For each pair of indices $(i, t)$, its associated data is given by the observation-action-reward triplet $(O_{i,t}, A_{i,t}, R_{i,t})$. In healthcare applications, each trajectory represents an individual patient where $O_{i,t}$ denotes the covariates of the $i$-th patient at time $t$, $A_{i,t}$ denotes the treatment assigned to the patient, and $R_{i,t}$ measures their clinical outcome at time $t$.

We investigate a confounded setting characterized by the presence of unmeasured confounders (denoted by $Z_{i,t}$) that influence both $A_{i,t}$ and $(R_{i,t}, O_{i,t+1})$ for each pair $(i, t)$. The offline data generating process (DGP) can be described as follows: (1) At each time $t$, the observation $O_{i,t}$ is recorded for the $i$-th trajectory. (2) Subsequently, an action $A_{i,t}$ is assigned for the $i$-th subject according to a behavior policy $\pi_b$, such that $A_{i,t} \sim \pi_b(\bullet|O_{i,t}, Z_{i,t})$. (3) Next, we obtain the immediate reward $R_{i,t}$ and the next observation $O_{i,t+1}$ such that $(R_{i,t}, O_{i,t+1}) \sim \mathcal{P}(\bullet|A_{i,t}, O_{i,t}, Z_{i,t})$ for some transition function $\mathcal{P}$. (4) Steps 2 and 3 are repeated until we reach the termination time $T$. See Figure 1(a) for an illustration.

In contrast, following a given target policy $\pi$ we wish to evaluate, the data is generated as follows: (1) At each time $t$, the action $A_{i,t}$ is determined by the target policy $\pi(\bullet|O_{i,t})$, independent of the unmeasured confounder $Z_{i,t}$. (2) The immediate reward $R_{i,t}$ and next observation $O_{i,t+1}$ are generated according to the transition function $\mathcal{P}(\bullet|A_{i,t}, O_{i,t}, Z_{i,t})$. In this setup, the unmeasured confounders affect only the reward and next observation distributions, but not the action. This is a primary difference from the offline data generating process. Specifically, whatever relationship exists between the unmeasured confounders and the actions in the offline data, that relationship is no longer in effect when we perform the target policy $\pi$. Our objective lies in evaluating the expected cumulative reward under $\pi$, given by

$$\eta^\pi = \frac{1}{NT} \sum_{i=1}^{N} \sum_{t=1}^{T} \mathbb{E}^\pi(R_{i,t}),$$

where $\mathbb{E}^\pi(R_{i,t})$ represents the expectation under the target policy $\pi$, irrespective of the unmeasured confounders.

To better understand the proposed two-way unmeasured confounding assumption, we first introduce two alternative assumptions concerning the unmeasured confounders as follows:

(a) *Unconstrained unmeasured confounding* (UUC): Each pair of indices $(i, t)$ corresponds to an unmeasured confounder $Z_{i,t}$, and there are no restrictions on these values.

(b) *One-way unmeasured confounding* (OWUC): For any $i$, the unmeasured confounders remain the same across time, i.e., $H_i = Z_{i,1} = Z_{i,2} = \cdots = Z_{i,T}$.

These two assumptions represent two extremes. The UUC assumption offers maximal flexibility without imposing any specific conditions. In contrast, the OWUC assumption is restrictive, essentially excluding 'time-varying confounders'.

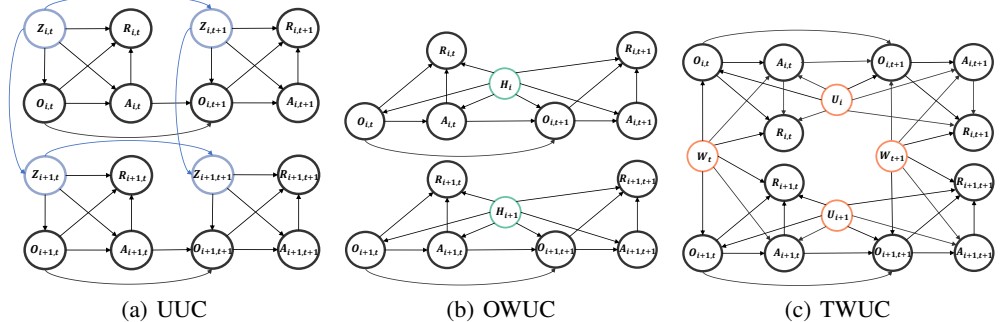

| (a) UUC | (b) OWUC | (c) TWUC |

Figure 1: The directed acyclic graphs of data generating processes under different assumptions. $(a) : \{z_{i,t}\}_{i,t}$ (colored in blue) are unconstrained unmeasured confounders. $(b) : \{h_i\}_i$ (colored in green) are one-way unmeasured confounders. $(c) : \{u_i\}_i$ and $\{w_t\}_t$ (colored in orange) are two-way unmeasured confounders.

The validity of the deconfounder algorithm [Wang and Blei, 2019] relies crucially on the consistent[3] estimation of the latent confounders. This is because it employs a plug-in method for constructing the average treatment effect estimator, which plugs in the estimated latent confounders into the model.

Unconstrained unmeasured confounding imposes no restrictions on the latent variables, but requires to estimate a total of $NT$ latent variables, which is equal to the sample size. This make consistent estimation infeasible without resorting to the 'deterministic unmeasured confounding' assumption discussed in Section 1. On the other hand, one-way unmeasured confounding only requires estimating $N$ latent variables, but in reality, this assumption is often difficult to meet.

In this article, we propose the following two-way unmeasured confounding, which offers a middle ground between the UUC assumption and the OWUC assumption:

(c) *Two-way unmeasured confounding* (TWUC): There exist time-invariant confounders $\{U_i\}_i$ and trajectory-invariant confounders $\{W_t\}_t$ such that $Z_{i,t} = (U_i^\top, W_t^\top)^\top$.

As discussed in the introduction, TWUC requires that all unmeasured confounders belong to one of two groups: trajectory-specific time-invariant confounders and time-specific trajectory-invariant confounders. Notably, it excludes confounders that are both trajectory- and time-specific. The $U_i$s can be interpreted as individual baseline information (e.g., salary or educational background) that remains consistent over time, while the $W_t$s represent external factors (e.g., weather or holidays) exerting a common influence across all trajectories. This assumption effectively relaxes one-way unmeasured confounding by accommodating time-varying confounders. Meanwhile, the number of latent confounders is confined to $N + T$, much smaller than the sample size $N \times T$ when both $N$ and $T$ grow to infinity. This ensures the feasibility of consistent estimation. See Figure 1 for a graphical visualization of the three assumptions.

To further elaborate the three modeling assumptions (a) – (c), we consider a linear model setup where the conditional means of the next observation and the immediate reward are linear functions of the current observation-action pair as well as the unmeasured confounders, and summarize the implications of adopting the three assumptions in the following corollaries.

**Proposition 1** (Inconsistency of the unconstrained model). The least square estimator (LSE) based on the UUC assumption cannot yield consistent predictions. Its mean square error (MSE) remains constant as both $N$ and $T$ increase.

**Proposition 2** (Inconsistency of the one-way model under misspecification). When time-varying unmeasured confounders exist, the LSE based on the one-way model cannot yield consistent predictions. Its MSE remains constant as both $N$ and $T$ increase.

**Proposition 3** (Consistency of the two-way model). The LSE based on the two-way model can yield consistent predictions. Its MSE decays to zero as both $N$ and $T$ increase.

Furthermore, we design a linear simulation setting to numerically compare the estimators based on these three assumptions and report their MSEs in Figure 2(b). The results reveal that the unconstrained

---

[3]Here, consistency means that the estimators converge to their ground truth as the sample size increases [see e.g., Casella and Berger, 2021].

model tends to overfit the data, as evidenced by its lowest prediction error on the training dataset and higher error in off-policy value estimation. Conversely, the one-way model underfits the data. It achieves the largest errors in both training data prediction and off-policy value estimation. In contrast, our proposed two-way model strikes a balance, resulting in the lowest error in off-policy value estimation.

To conclude this section, we summarize the DGP under the proposed two-way unmeasured confounding assumption:

- At the initial time, each trajectory independent generates an observed $O_{i,1}$ and an unobserved $U_i$. Additionally, a latent $W_1$ is independently generated.
- Next, at each time $t$, $A_{i,t}$ and $(R_{i,t}, O_{i,t+1})$ are generated, according to $\pi_b$ (or a target policy $\pi$) and $\mathcal{P}$, respectively. Moreover, a latent $W_{t+1}$ is generated, whose distribution depends only on the past time-specific confounders.

Under this DGP, the two-way unmeasured confounders are policy-agnostic, i.e., unaffected by the actions or policies. Consequently, we can employ a plug-in approach to construct the policy value estimator (see (1)), eliminating the need to estimate their distributions.

# 3 Two-way Deconfounder

We introduce the proposed deconfounder algorithm in this section. We first present the proposed neural network architecture under two-way unmeasured confounding. We next define the loss function used for model training. Finally, we introduce a model-based policy value estimator, built upon the estimated model.

**Network Architecture.** The proposed model contains the following components: (1) $d$-dimensional embedding vectors $\{u_i\}_i$ to model the trajectory-specific latent confounders; (2) $d$-dimensional embedding vectors $\{w_t\}_t$ to model the time-specific latent confounders; (3) A transition function $\widehat{\mathcal{P}}(\bullet|a,o,u_i,w_t)$ that takes an action-observation pair $(a,o)$ and a pair of embedding vectors $(u_i,w_t)$ as input to model the conditional distribution of the reward-next-observation pair given the current observation-action pair and the two-way unobserved confounders; (4) An actor network $\widehat{\pi}_b(\bullet|o,u_i,w_t)$ that takes $o$ and $(u_i,w_t)$ as input to model the behavior policy.

Our objective is to simultaneously learn the embedding representations and the parameters in both the transition and actor networks from the observed data. Toward that end, we treat each pair of embedding vectors $(u_i,w_t)$ as two entities. To accurately capture their joint effects on both the transition function and the behavior policy, we adopt the neural tensor network [NTN, Socher et al., 2013], which is known for its ability to capture the intricate interactions between pairs of entity vectors.

Specifically, we parameterize $\widehat{\mathcal{P}}(\bullet|a,o,u_i,w_t)$ via a conditional Gaussian model given by $\mathcal{N}(\widehat{\mu}(a,o,u_i,w_t),\mathrm{diag}(\widehat{\sigma}(a,o,u_i,w_t)))$, where $\mathrm{diag}(\widehat{\sigma}(a,o,u_i,w_t))$ is a diagonal matrix and $\widehat{\sigma}(a,o,u_i,w_t)$ denotes the vector consisting of all diagonal elements. Its conditional mean and variance functions are modeled jointly with the behavior policy, specified by

$$(\widehat{\mu}^\top,\widehat{\sigma}^\top)^\top = \mathrm{MLP}_{\mathcal{P}}(a_{i,t},\mathrm{NTN}(o,u_i,w_t)),\ \widehat{\pi}_b(\bullet\mid o,u_i,w_t) = \mathrm{MLP}_{\pi_b}(\mathrm{NTN}(o,u_i,w_t)),$$

where $\mathrm{NTN}(o,u_i,w_t)$ denotes the output vector from an NTN, and $\mathrm{MLP}_{\mathcal{P}}$, $\mathrm{MLP}_{\pi_b}$ are the two multilayer perceptrons that take this output vector as input. The detailed architecture of NTN layer is presented in Appendix B.1. As such, the NTN layer captures the joint information from both the observation $o$ and latent confounders $u_i$, $w_t$ and is shared among the actor and transition networks. We provide an overview of the proposed model architecture in Figure 2 (a).

Finally, we remark that while we model the transition function using a conditional Gaussian like other works in the RL literature [see e.g., Janner et al., 2019, Yu et al., 2020], more complex generative models such as transformers or diffusion models are equally applicable.

**Loss Function**. As mentioned earlier, the proposed model contains both the transition network and the actor network. Accordingly, our loss function is given by

$$L(\mathcal{D};\{u_i\}_i,\{w_t\}_t) = (1-\alpha)\cdot L_T + \alpha\cdot L_A,$$

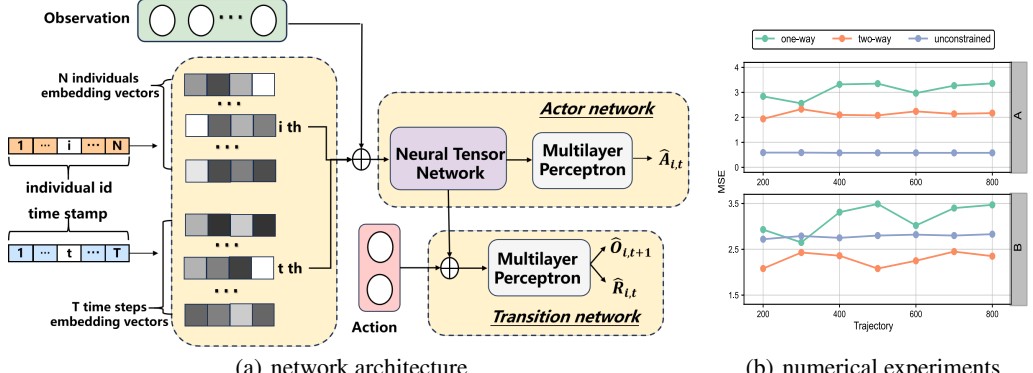

(a) network architecture          (b) numerical experiments

Figure 2: $(a)$ : An overview of the proposed network architecture. $(b)$ : The upper panel reports MSEs under different unmeasured confounding assumptions for fitting the observed data whereas the bottom panel displays the MSEs for off-policy value prediction. The unconstrained unmeasured confounding model shows the best fit for the training data, due to overfitting. The OPE estimator under the proposed two-way unmeasured confounding achieves the smallest MSE. More details are referred to Appendix D.1.

where $L_T$ is the negative log likelihood of conditional Gaussian model for the transition network, $L_A$ is the cross-entropy loss between the actual action and actor network, and $\alpha \in (0, 1)$ is a hyperparameter that balances the two losses. This loss is optimized to compute both the latent embeddings vectors $(u_i, w_t)$s and the parameters in the three neural networks: NTN, $\mathrm{MLP}_{\mathcal{P}}$ and $\mathrm{MLP}_{\pi_b}$. Further details are relegated to Appendix C.1 to save space.

Alternative to our loss function which involves both the actor and transition networks, one can consider minimizing other losses focused exclusively on either the transition or the actor network, but not necessarily both. However, in the presence of unmeasured confounding, it is essential to note that both the behavior policy and transition function are influenced by these latent confounders. Consequently, our joint learning approach is expected to more effectively identify the unmeasured confounders compared to these transition-only or actor-only approaches.

**Model-based OPE.** Based on the estimated model, we develop a model-based estimator to learn $\eta^\pi$ via Monte Carlo policy evaluation [Sutton and Barto, 2018]. A key observation is that, since the two-way unmeasured confounders are policy-agnostic, the empirical sum shown below is an unbiased intermediate estimator for the evaluation target $\eta^\pi$:

$$\frac{1}{NT} \sum_{i=1}^{N} \sum_{t=1}^{T} \mathbb{E}^\pi \Big( R_{i,t} \mid O_{i,1}, U_i, \{W_{t'}\}_{t'=1}^{t} \Big). \tag{1}$$

Note that in this formulation, the latent factors in the conditioning set can be substituted with our estimated embedding vectors. Additionally, based on our estimated transition network, the expectation $\mathbb{E}^\pi$ can be effectively approximated using the Monte Carlo method. This approach allows us to construct a model-based plug-in estimator for Equation (1).

To elaborate, the Monte Carlo simulation begins at $t = 1$ for each individual $i$. We sample an action $\widehat{A}_{i,t}$ according to the target policy $\pi(\bullet \mid \widehat{O}_{i,t})$, draw samples for $\widehat{R}_{i,t}$ and $\widehat{O}_{i,t+1}$ from the learned transition network $\widehat{\mathcal{P}}(\bullet \mid \widehat{A}_{i,t}, \widehat{O}_{i,t}, \widehat{U}_i, \widehat{W}_t)$ with the estimated latent factors $\{\widehat{U}_i\}_i$, $\{\widehat{W}_t\}_t$, and iterate this procedure until the terminal time $T$ is reached. Next, we replicate this simulation multiple times to reduce the Monte Carlo error. The final step is to aggregate all the estimated results across these simulations to construct the final OPE estimator.

## 4 Theoretical Results

In this section, we provide a finite-sample error bound for the expected difference between the estimated policy value and the ground truth $\eta^\pi$. We first introduce some notations. We consider a tabular setting where the observation space $\mathcal{O}$, action space $\mathcal{A}$ and latent factor spaces $\mathcal{U}$, $\mathcal{W}$ are all discrete. Let $d_{\bar{\mathcal{D}}}$ denote the data distribution of quadruples $(a, o, u, w)$, $d_{\bar{\mathcal{D}}}(a, o, u, w) = (NT)^{-1} \sum_{(A,O,U,W) \in \bar{\mathcal{D}}} \mathbb{P}(A = a, O = o, U = u, W = w)$, where the summation $\sum_{(A,O,U,W) \in \bar{\mathcal{D}}}$

is carried out over all quadruples in the augmented dataset $\bar{\mathcal{D}} = \mathcal{D} \cup \{U_i\}_i \cup \{W_t\}_t$ containing both the observed data and the latent confounders. Furthermore, let $d^\pi$ represent the visitation distribution under $\pi$. Specifically, $d^\pi(a, o, u, w)$ denotes the average probability of a given quadruple $(a, o, u, w)$ appearing at any time step under $\pi$ (refer to Appendix B.1 for the detailed definition).

We next introduce some assumptions to derive the finite-sample error bound of the proposed policy value estimator.

**Assumption 1** (Coverage). The data distribution covers the visitation distribution induced by the target policy $\pi$, i.e., $C = \sup_{a,o,u,w} \frac{d^\pi(a,o,u,w)}{d_{\bar{\mathcal{D}}}(a,o,u,w)} < \infty$.

**Assumption 2** (Boundedness). The absolute values of the immediate reward are upper bounded by some constant $R_{\max} < \infty$.

**Assumption 3** (Error bound of estimated transition function). $\mathbb{E}\|\widehat{\mathcal{P}} - \mathcal{P}\|_{d_{\bar{\mathcal{D}}}}^2 \leq \varepsilon_{\mathcal{P}}^2$ for some $\varepsilon_{\mathcal{P}} > 0$ where $\|\widehat{\mathcal{P}} - \mathcal{P}\|_{d_{\bar{\mathcal{D}}}}$ denotes the total variation distance between $\widehat{\mathcal{P}}$ and $\mathcal{P}$ (see Appendix B.1 for the detailed definition).

**Assumption 4** (Error bound of estimated latent confounders). $\sum_{1 \leq i \leq N} \mathbb{E}\|\widehat{U}_i - U_i\|_2^2 / N \leq \varepsilon_{U,W}^2$ and $\sum_{1 \leq t \leq T} \mathbb{E}\|\widehat{W}_t - W_t\|_2^2 / T \leq \varepsilon_{U,W}^2$ for some $\varepsilon_{U,W} > 0$.

**Assumption 5** (Autocorrelation). For any $t_1, t_2 \geq 1$, let $\rho(t_1, t_2)$ denote the correlation coefficient between $\mathbb{E}^\pi(R_{1,t_1} | \{W_{t'}\}_{t'=1}^{t_1})$ and $\mathbb{E}^\pi(R_{1,t_2} | \{W_{t'}\}_{t'=1}^{t_2})$. There exists some $0 \leq \alpha \leq 1$ such that $\sum_{1 \leq t_1 \neq t_2 \leq T} \rho(t_1, t_2) = O(T^{2\alpha})$.

We remark that conditions similar to Assumptions 1-2 are widely imposed in the RL literature to simplify the theoretical analysis [see e.g., Chen and Jiang, 2019, Fan et al., 2020, Liu et al., 2020, Uehara and Sun, 2021]. Assumption 3 is concerned with the estimation error of the transition function. This error is expected to be minimal, since we use neural networks for function approximation [see e.g., Schmidt-Hieber, 2020, Farrell et al., 2021]. Assumptions 4 is concerned with the estimation errors of the estimated two-way unmeasured confounders. According to Proposition 3, these errors are negligible under simple models. Finally, Assumption 5 is purely technical. It measures the autocorrelation of the time series $\mathbb{E}^\pi(R_{1,t} | \{W_{t'}\}_{t'=1}^t)$. Here, $\alpha = 0$ indicates independence over time. This condition is automatically satisfied when $\alpha = 1$.

**Theorem 1** (Finite-sample error bound). Suppose the two-way unmeasured confounding assumption holds, and Assumptions 1-5 are satisfied. Then

$$\mathbb{E}|\widehat{\eta}^\pi - \eta^\pi| \leq CTR_{\max}\varepsilon_{\mathcal{P}} + cTR_{\max}\varepsilon_{U,W} + cR_{\max}N^{-1/2} + cR_{\max}T^{\alpha-1},$$

for some constant $c > 0$.

It can be seen from Theorem 1 that the mean absolute error of the proposed policy value estimator involves two components:

1. **Estimation errors:** The first two terms on the right side of the inequality correspond to the estimation errors of the transition function and latent confounders respectively. Notably, both terms are linear in the time horizon $T$, due to error accumulation (see Appendix B.5). However, such a linear dependence is the best one can hope in general [Jiang, 2024], although it is possible to eliminate the dependence upon $T$ under additional ergodicity assumptions [Liao et al., 2022].

2. **Standard deviations:** The last two terms measure the standard deviations of the average values across trajectories and over time, respectively. These upper bounds decays to zero, as both $N$ and $T$ approach infinity, provided that the exponent $\alpha$ in Assumption 5 – which measures the autocorrelation of the time series $\mathbb{E}^\pi(R_{1,t} | \{W_{t'}\}_{t'=1}^t)$ – is strictly smaller than 1.

## 5 Experiments

In this section, we perform numerical experiments using two simulated datasets and one real-world dataset to demonstrate the effectiveness of the proposed two-way deconfounder (denoted by TWD) in handling unmeasured confounders. We consider two simulated examples in Section 5.1: a simple dynamic process and a tumor growth example. For each simulated example, the true value of $\eta^\pi$ is computed based on 10,000 Monte Carlo experiments. We also explore a real-world example using the

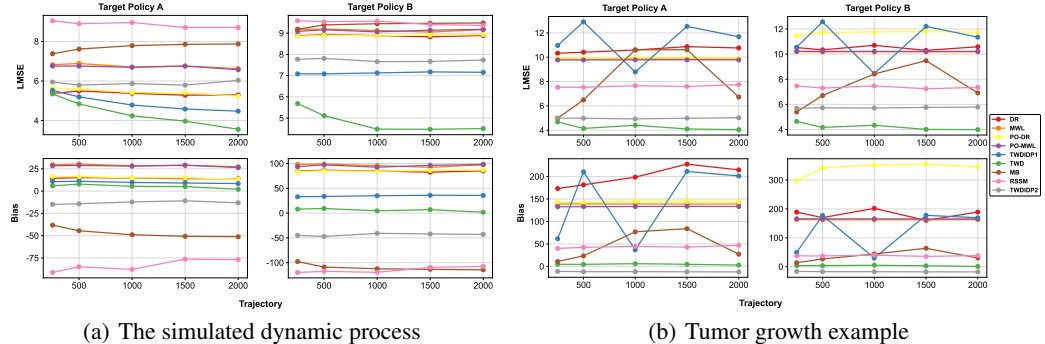

(a) The simulated dynamic process  (b) Tumor growth example

Figure 3: Logarithmic MSE and Bias of various estimators for the simulated dynamic process and tumor growth example.

MIMIC-III dataset in Section 5.2, and conduct a sensitivity analysis and an ablation study in Sections 5.3 and 5.4, respectively. The source code is available on Github: https://github.com/fsmiu/Two-way-Deconfounder.

We use two metrics to evaluate different OPE estimators: the logarithmic mean squared error (LMSE) and bias. Both are estimated based on 20 simulations. Comparison is made between TWD and the following set of baseline methods, which covers a wide range of model-based and model-free approaches:

(1) **Model-based method** (MB) that learns a transition model from the offline data based on the proposal by Yu et al. [2020] and applies the Monte Carlo method to construct the OPE estimator;

(2) **Minimax weight learning** [MWL, Uehara et al., 2020] that learns a marginalized importance sampling (MIS) ratio from the offline data to constructs an MIS estimator for OPE;

(3) **Double robust method** (DR) that combines the MIS ratio and an estimated Q-function computed via minimax learning [Uehara et al., 2020] to enhance robustness of OPE;

(4) **Partially observable MWL** [PO-MWL, Shi et al., 2022a] – a POMDP-type method that extends MWL to handle unmeasured confounders;

(5) **Partially observable DR** [PO-DR, Shi et al., 2022a] – another POMDP-type method that extends DR to handle unmeasured confounders;

(6) **Recurrent state-space method** [RSSM, Hafner et al., 2019a,b] that models unmeasured confounders as latent states;

(7) **Model-free two-way doubly inhomogeneous decision process** (TWDIDP1) – a deconfounding-type method that uses the model-free algorithm developed by Bian et al. [2023];

(8) **Model-based two-way doubly inhomogeneous decision process** (TWDIDP2) – another model-based deconfounding-type algorithm, also developed by Bian et al. [2023].

Notably, the first three methods require the NUC assumption. The next three methods are POMDP-type, whereas the last two are deconfounding-type algorithms. Given our focus on settings without external proxies, we do not compare against methods developed under memoryless unmeasured confounding, which typically rely on these proxies, as commented in Section 1.

### 5.1 Simulation studies

**Simulated Dynamic Process**. We first consider a dynamic process with four-dimensional observations and binary actions. The data is generated under the proposed TWUC assumption; see Appendix D.2 for its detailed DGP. We fix $T = 50$, and vary the number of trajectories from 250 to 2000. As shown in Figure 3(a), our proposed TWD estimator frequently achieves the smallest LMSE with bias closer to 0 in all cases. Additionally, the LMSE of TWD generally decreases as the number of trajectories increases, demonstrating its consistency. In contrast, most other methods under the assumption of NUC or POMDP setting are severely biased, highlighting the risks of ignoring or improperly handling unmeasured confounding.

**Tumor Growth Example**. We consider a tumor growth example and utilize the pharmacokinetic-pharmacodynamic (PK-PD) model for data generation; see Appendix D.3 for details. The observation

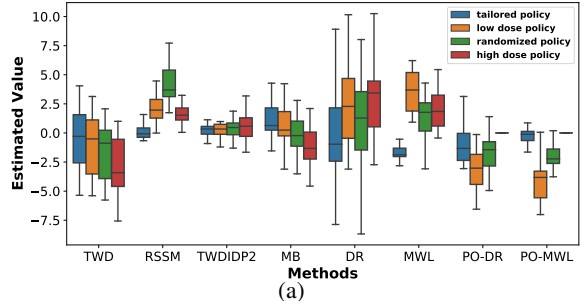

| | Number of trajectories | |
|---|---|---|
| Model | 2000 | 3708 |
| MB | 0.255±0.008 | 0.252±0.005 |
| RSSM | 0.332±0.014 | 0.329±0.008 |
| TWDIDP2 | 0.262±0.008 | 0.259±0.006 |
| TWD | **0.254±0.008** | **0.251±0.006** |

(b)

(a)

Figure 4: $(a)$ : The estimated policy value for four target policies in real-world dataset. $(b)$ : Average root MSE and its standard error in the results for predicting immediate reward and next observation. The results are aggregated over 20 runs.

is two-dimensional, including the tumor volume $V(t)$ and the chemotherapy drug concentration $C(t)$. The action space $\mathcal{A}$ includes two treatments: radiotherapy $A^r(t)$ and chemotherapy $A^c(t)$. We evaluate two target policies: a random policy (denoted by 'A') and a individualized policy that is tailored to the patients' conditions (denoted by 'B'). As shown in Figure 3(b), TWD consistently achieves the lowest LMSE and is empirically unbiased in all cases, demonstrating the effectiveness of our method. In contrast, other methods yield biased estimates, resulting in significantly higher LMSEs. Importantly, the performance of these alternative methods does not show significant improvement with an increase in the total number of trajectories. These results underscore the applicability of our method in a more realistic scenario.

## 5.2 Real-world example: MIMIC-III database

In this section, we apply the proposed two-way deconfounder to the medical information mart for intensive care (MIMIC-III) database [Johnson et al., 2016]. We extract 3,707 patients with trajectories up to 20 timesteps. Following the analysis of Zhou et al. [2023b], we define a $5 \times 5$ action space and set the reward to the difference between current SOFA score and next SOFA score, so a lower reward indicates a higher risk of mortality. We also extract 12 covariates as the observation; further details are described in Appendix D.4. The dataset is likely non-stationary and lacks patient's personal information. Therefore, it might be reasonable to employ TWUC to model the unmeasured confounders [Bian et al., 2023].

Given that this is a real dataset, we do not have access to the true value of the target policy. To compare TWD against other baselines, we employ two approaches. The first approach uses 90% of data for training, and the remaining 10% for evaluating an algorithm's prediction error for the reward and next observation. Notice that this approach is applicable to evaluate model-based methods only. We report all mean squared prediction errors in Figure 4(b). It can be seen that TWD results in the lowest prediction errors in all cases, demonstrating its effectiveness to infer unobserved confounders.

The second approach assesses each algorithm ability in distinguishing between tailored individualized policies and other random, non-individualized policy. An effective OPE algorithm should consistently rank an individualized policy as the superior policy. In what follows, four policies are evaluated using TWD and other baseline methods: a randomized policy, a non-individualized high dose policy, a non-individualized low dose policy and an tailored individualized policy. As shown in Figure 4(a), TWD and MB can effectively distinguish the individualized policy from other policies, with the individualized policy consistently achieving the highest estimated value. However, other methods lead to strange conclusions. For example, the result from TWDIDP2 suggest that all policies achieve similar values. Consequently, results from MWL,DR, PO-DR and RSSM suggest that the high dose policy is better than the tailored individualized policy. Additionally,TWDIDP1 performs specially poor, rendering it unsuitable for display alongside other methods in this figure.While these results require further validation by medical professionals, they highlight the potential of the proposed method in real-world medical applications.

## 5.3 Sensitivity analysis

In this section, we investigate how TWD performs when the proposed TWUC assumption is violated. Specifically, we vary unmeasured confounders that are both trajectory- and time-specific in the

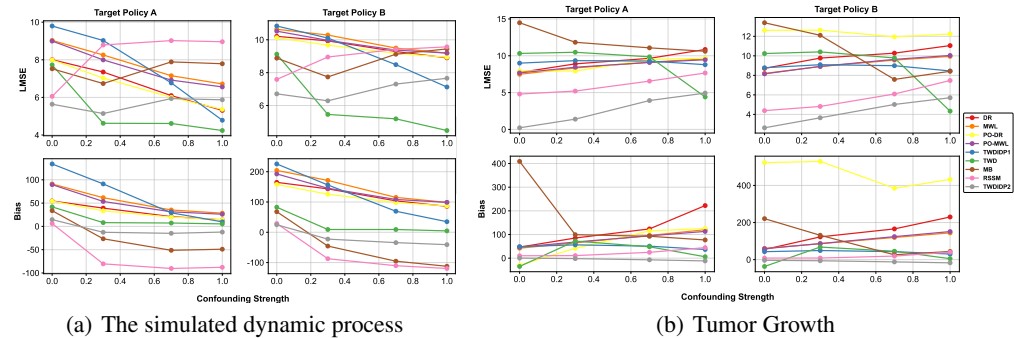

|            | Target Policy A | Target Policy B |            | Target Policy A | Target Policy B |
| (a) The simulated dynamic process | | | (b) Tumor Growth | | |

Figure 5: Sensitivity analysis for the simulated dynamic process and tumor growth experiment.

reward function and behavior policy and we introduce a sensitivity parameter $\Gamma$ to quantify the extent to which the proposed TWUC assumption is violated. When $\Gamma = 1$, the proposed TWUC assumption holds; as $\Gamma$ decreases towards zero, the assumption is increasingly violated. We vary $\Gamma = \{0.0, 0.3, 0.7, 1.0\}$ in the experiments, and fix the number of the trajectory to 1000. See further details in Appendix D.5.

We focus on the two simulated examples. As shown in Figure 5(a), in the simulated dynamic process, the performance of the proposed methods remains relatively stable as long as $\Gamma > 0$. However, when $\Gamma = 0$ – where the TWUC is completely violated – TWD loses its superiority. Meanwhile, as shown in Figure 5(b), in the tumor growth example, the performance of TWD is very sensitive to $\Gamma$, and TWD performs better than other methods only if $\Gamma = 1.0$.

## 5.4 Ablation study

We conduct an ablation study to compare TWD against the following variants:

(1) **TWD with transition-only loss function** (TWD-TO): This variant employs the proposed TWUC assumption, but removes the cross-entropy loss from the objective function. Consequently, it solely uses the transition model to learn the two-way embedding vectors, without modeling the behavior policy during training.

(2) **TWD without neural tensor network** (TWD-MLP): In this variant, the neural tensor network is replaced with an MLP.

(3) **One-way deconfounder without individual embedding** (OWD-NI): This variant removes the individual embedding vector, operating under the one-way unmeasured confounding assumption.

(4) **One-way deconfounder without time embedding** (OWD-NT): This variant removes the time embedding vector.

We fix the number of trajectories to 1000 and report the LMSEs of various estimators in Table 1. It can be seen that: (i) OWD-NI and OWD-NT significantly underperform TWD due to their reliance on the one-way unmeasured confounding assumption. (ii) TWD consistently outperforms TWD-MLP, due to the neural tensor network's ability to capture intricate interactions between the trajectory-specific and the time-specific unmeasured confounders. (iii) In the simulated dynamic process, TWD achieves better performance than TWD-TO. This could be attributed to our proposed joint learning strategy, which simultaneously estimates both the transition function and the behavior policy and enhances the model's capability to infer unmeasured confounders. However, TWD performs worse than TWD-TO in the tumor growth example.

**Table 1: Ablation study for variants of TWD**

|         | Environments | | | |
|         | DP | | TG | |
| **Model** | A | B | A | B |
| TWD     | **4.23** | **4.47** | 4.42 | 4.33 |
| TWD-TO  | 4.72 | 5.14 | **3.58** | **3.39** |
| TWD-MLP | 4.34 | 5.26 | 4.56 | 4.44 |
| OWD-NI  | 7.50 | 8.59 | 6.80 | 6.92 |
| OWD-NT  | 6.78 | 8.92 | 6.26 | 6.31 |

[1] DP: the simulated dynamic process, TG: the tumor growth example, A: Target policy A, B: Target policy B.

In summary, these results demonstrate the effectiveness of the proposed two-way unmeasured confounding model and our joint learning strategy, along with the crucial role of the neural tensor network in capturing complex interactions, all contributing to TWD's superior performance.

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

# A Discussion

## A.1 Critiques of Wang and Blei [2019]

In this section, we discuss the critiques of the paper Wang and Blei [2019] and illustrate how our proposal addresses these criticisms.

Specifically, Wang and Blei [2019] imposed the "consistency of substitute confounders" assumption which requires that the unmeasured confounders $Z_i$ can be consistently estimated from the causes $A_i$. However, this assumption indeed invalidates the derivations of Theorems 6-8 of Wang and Blei [2019], resulting in the inconsistency of the algorithm. Specifically, as commented by D'Amour [2019], if the event $A = a$ provides a perfect measurement of $Z$ such that there is some function $\hat{z}(A)$ such that $\hat{z}(a) = Z$, then the overlap condition fails. As a result, the ATE cannot be consistently identified. Ogburn et al. [2019] expressed similar concerns towards the algorithm.

Unlike Wang and Blei [2019], our algorithm does not require such an assumption. Under the RL setting, the proposed two-way unmeasured confounding assumption effectively limits the number of unmeasured confounders to $\mathcal{O}(N) + \mathcal{O}(T)$, which facilitates their consistent estimation when both the number of trajectories $N$ and the number of decision points per trajectory $T$ grow to infinity, avoiding the need for the unmeasured confounders to be deterministic functions of the actions.

## A.2 Limitations

Although the proposed two-way unmeasured confounding assumption is more flexible than the existing one-way unmeasured confounding, it might not adequately handle more complex scenarios involving latent confounders that are both trajectory- and time-specific, or policy-dependent. In practice, this assumption can be partially evaluated by testing whether including the estimated two-way confounders in the observation leads to a Markovian system [see e.g., Chen and Hong, 2012, Shi et al., 2020, Zhou et al., 2023c].

## A.3 Future work

Our proposal requires the existence of an observation which, along with the two-way confounders, blocks all backdoor paths between the treatment and both the immediate reward and future observations. Naïvely using pre-treatment variables as observations might lead to biased estimators in certain causal models like the M-graph Cinelli et al. [2022]. While confounder selection algorithms [Guo et al., 2023] are designed to address the problem, they mainly focus on the bandit setting. Our future work is to extend these algorithms to RL to further improve the practical applicability of our method.

# B Proofs

## B.1 Notations

We first introduce list the notations that will be used in our proof.

- $\text{NTN}(o, u_i, w_t) = f(g(u_i^\top W^{[1:k]} w_t + M\tau_{i,t} + b))$, where $\tau_{i,t}$ is shorthand for $[o_{i,t}^\top, u_i^\top, w_t^\top]^\top$, $f$ is a standard linear layer and $g$ is the activation function. Within this function, $W^{[1:k]} \in \mathbb{R}^{d \times d \times k}$ is a three-dimensional tensor and the bilinear tensor products $u_i^T W^{[1:k]} w_t$ result in a vector whose $m$-th entry is given by $u_i^T W^{[m]} w_t$, where $W^{[m]}$ denotes the $m$-th slice of the tensor. $M \in \mathbb{R}^{k \times (d_o + 2d)}$, where $d_o$ represents the dimension of observation $o$, and $b \in \mathbb{R}^k$ is the bias term.

- $R(a, o, u, w) := \sum_{r,o'} r\mathcal{P}(r, o'|a, o, u, w)$, which denotes the expectation of reward given observation $o$, action $a$ and latent factors $(u, w)$, and $\widehat{R}$ denotes its estimator.

- $P(o'|a, o, u, w) := \sum_r \mathcal{P}(r, o'|a, o, u, w)$ denotes the observation transition function, which calculates the probability of transitioning from observation $o$ to observation $o'$ given the action $a$ and latent factors $(u, w)$, and $\widehat{P}$ denotes its estimator.

- $\rho_0(\bullet)$ denotes the probability mass function of $(O_{i,1}, U_i)$.

- $d_{\bar{\mathcal{D}}}$ denotes the data distribution of quadruples $(a, o, u, w)$, given by $d_{\bar{\mathcal{D}}}(a, o, u, w) = T^{-1} \sum_{t=1}^T \mathbb{P}(O_{1,t} = o, A_{1,t} = a, U_1 = u, W_t = w)$.

- $d^\pi$ denotes the distribution of quadruples $(a, o, u, w)$ in a trajectory of horizon $T$ generated under $\pi$, i.e., $d^\pi(a, o, u, w) = T^{-1} \sum_{t=1}^T \mathbb{P}^\pi(O_{1,t} = o, A_{1,t} = a, U_1 = u, W_t = w)$.

- $\text{TV}(\widehat{\mathcal{P}}(\bullet|a, o, u, w), \mathcal{P}(\bullet|a, o, u, w)) := \sum_{r,o'} |\widehat{\mathcal{P}}(r, o'|a, o, u, w) - \mathcal{P}(r, o'|a, o, u, w)|/2$ denotes the total variation between $\widehat{\mathcal{P}}(\bullet|a, o, u, w)$ and $\mathcal{P}(\bullet|a, o, u, w)$.

- $\|\widehat{\mathcal{P}} - \mathcal{P}\|_{d_{\bar{\mathcal{D}}}} := \sqrt{\mathbb{E}_{(A,O,U,W) \sim d_{\bar{\mathcal{D}}}} \text{TV}^2(\widehat{\mathcal{P}}(\bullet|A, O, U, W), \mathcal{P}(\bullet|A, O, U, W))}$ is used to measure the average estimation error of $\widehat{\mathcal{P}}$ on the given offline dataset.

Additionally, we will use a generic constant $c$ in the proof, whose value is allowed to vary from place to place.

## B.2 Proof of Proposition 1

*Proof.* Notice that the MSE of the predicted reward-next-observation pair is equal to the sum of the MSE of the predicted immediate reward and that of the predicted next observation. Consequently, it suffices to show the MSE of the predicted immediate reward remains constant.

Toward that end, we consider the following linear model under unconstrained unmeasured confounding,

$$R_{i,t} = (O_{i,t}, A_{i,t})\zeta + Z_{i,t} + \varepsilon_{i,t},$$

where $Z_{i,t}$s are one-dimensional, $\zeta = (\zeta_1, \zeta_2)^\top = \mathbb{R}^2$ denotes the coefficient vector, and $\varepsilon_{i,t}$ are mean zero i.i.d. measurement errors with each $\varepsilon_{i,t}$ being independent of the set of variables $\mathcal{H}_{i,t} = \{(O_{i',t'}, A_{i',t'}, Z_{i',t'}) : i' \neq i \text{ or } t' \leq t\}$.

In vector and matrix notations, we define $Y = (R_{1,1}, \ldots, R_{1,T}, \ldots R_{N,2}, \ldots, R_{N,T})^\top$ as the $(NT)$-dimensional outcome vector, $\beta = (\zeta_1, \zeta_2, Z_{1,1}, \ldots, Z_{1,T}, \ldots, Z_{N,1}, \ldots, z_{N,T})^\top$ as the $(NT + 2)$-dimensional coefficient vector, $\mathcal{E} = (\varepsilon_{1,1}, \ldots, \varepsilon_{1,T}, \ldots, \varepsilon_{N,1}, \ldots, \varepsilon_{N,T})^\top \in \mathbb{R}^{NT}$ as the set of residuals and

$$X = \begin{pmatrix} O_{1,1} & A_{1,1} & 1 & 0 & \cdots & 0 \\ O_{1,2} & A_{1,2} & 0 & 1 & \cdots & 0 \\ \cdots & \cdots & \cdots & \cdots & \ddots & \cdots \\ O_{N,T} & A_{N,T} & 0 & 0 & \cdots & 1 \end{pmatrix} \in \mathbb{R}^{NT \times (NT+2)}.$$

Thus, we can formulate the linear model as $Y = X\beta + \mathcal{E}$. Since the LSE $\widehat{\beta}$ is an unbiased estimator, its predictor $\widehat{Y} = X\widehat{\beta}$'s MSE can be calculated as follows,

$$\begin{aligned} \text{MSE}(\widehat{Y}) &= \frac{1}{NT}\mathbb{E}\|\widehat{Y} - \mathbb{E}(Y|X\beta)\|_2^2 = \frac{1}{NT}\mathbb{E}\|X(X^\top X)^- X^\top \mathcal{E}\|_2^2 \\ &= \frac{1}{NT}\mathbb{E}[\mathcal{E}^\top X(X^\top X)^- X^\top \mathcal{E}] = \frac{1}{NT}\mathbb{E}\Big\{\text{trace}[X(X^\top X)^- X^\top \mathcal{E}\mathcal{E}^\top]\Big\} \qquad (2) \\ &= \frac{\sigma^2}{NT}\mathbb{E}\Big\{\text{trace}[X(X^\top X)^- X^\top]\Big\} = \sigma^2, \end{aligned}$$

where the conditional expectation $\mathbb{E}(Y|X\beta)$ is taken in a componentwise manner, $(X^\top X)^-$ denotes the generalized inverse of $X^\top X$ and $\sigma^2 = \text{Var}(\varepsilon_{i,t})$. Here, the second last equation holds due to the independence between $\varepsilon_{i,t}$ are $\mathcal{H}_{i,t}$. Consequently, the MSE is a fixed constant. This completes the proof. $\square$

## B.3 Proof of Proposition 2

*Proof.* Similar to the proof of Proposition 1, it suffices to prove that the MSE of the predicted immediate reward remains a constant. Toward that end, we assume the immediate reward satisfies the following linear two-way unmeasured confounding assumption,

$$R_{i,t} = (O_{i,t}, A_{i,t})\zeta + U_i + W_t + \varepsilon_{i,t}.$$

However, one-way unmeasured confounding assumes only the existence of trajectory-specific unmeasured confounders and ignores time-specific confounders, leading to the following model

$$R_{i,t} = (O_{i,t}, A_{i,t})\zeta + H_i + \varepsilon_{i,t}.$$

Because we are more concerned with estimating fixed effects and also to simplify subsequent symbols, we consider a scenario that is easier to estimate. Let $\widehat{\zeta}$ denote the LSE of $\zeta$ based on the one-way model. Since $X^\top \widehat{Y} = X^\top X(X^\top X)^{-1} X^\top Y = X^\top Y$, the LSE of $H_i$ satisfies

$$\widehat{H}_i = \frac{1}{T}\sum_{t=1}^T [R_{i,t} - (O_{i,t}, A_{i,t})\widehat{\zeta}] = U_i + \frac{1}{T}\sum_{t=1}^T [W_t - (O_{i,t}, A_{i,t})(\widehat{\zeta} - \zeta)].$$

Therefore, the MSE of the predicted $\widehat{R}_{i,t} = (O_{i,t}, A_{i,t})\widehat{\zeta} + \widehat{H}_i$ is given by

$$\frac{1}{NT}\sum_{i,t}\mathbb{E}[(O_{i,t}, A_{i,t})\widehat{\zeta} + \widehat{H}_i - (O_{i,t}, A_{i,t})\zeta - U_i - W_t]^2 = \frac{1}{T}\sum_{t=1}^T \mathbb{E}(W_t - \frac{1}{T}\sum_{t'=1}^T W_{t'})^2.$$

For a stationary and ergodic time series $\{W_t\}_t$, $T^{-1}\sum_{t'=1}^T W_{t'}$ is to converge to $\mathbb{E}(W_t)$ as $T \to \infty$ and thus, the above equation is to converge to the variance of $W_t$. Consequently, unless each $W_t$ is degenerate, the MSE will not decay to zero. This completes the proof. $\square$

## B.4  Proof of Proposition 3

*Proof.* To simplify the proof, we only show the MSE of predicted immediate reward decays to zero as both $N$ and $T$ grow to infinity. The two-way model can be similarly expressed as $Y = X\beta + \mathcal{E}$, where $Y = (R_{1,1}, \ldots, R_{1,T}, \ldots R_{N,1}, \ldots, R_{N,T})^\top \in \mathbb{R}^{NT}$, $\beta = (\zeta_1, \zeta_2, U_1, \ldots, U_N, W_1, \ldots, W_T)^\top \in \mathbb{R}^{N+T+2}$, $\mathcal{E} = (\varepsilon_{1,1}, \ldots, \varepsilon_{1,T}, \ldots \varepsilon_{N,1}, \ldots, \varepsilon_{N,T})^\top \in \mathbb{R}^{NT}$, and

$$
X = \begin{pmatrix}
O_1 & A_1 & \mathbf{1}_T & \mathbf{0}_T & \cdots & \mathbf{0}_T & I_T \\
O_2 & A_2 & \mathbf{0}_T & \mathbf{1}_T & \cdots & \mathbf{0}_T & I_T \\
\cdots & \cdots & \cdots & \cdots & \ddots & \cdots & \cdots \\
O_N & A_N & \mathbf{0}_T & \mathbf{0}_T & \cdots & \mathbf{1}_T & I_T
\end{pmatrix} \in \mathbb{R}^{NT \times (N+T+2)},
$$

where $\mathbf{1}_T = (1, \ldots, 1)^\top \in \mathbb{R}^T$, $\mathbf{0}_T = 0 \cdot \mathbf{1}_T$, and $I_T$ represents the $T \times T$ identity matrix.

Similar to Equation (2), the MSE of the predicted reward is equal to

$$
\mathrm{MSE}(\widehat{Y}) = \frac{1}{NT} \mathbb{E}[\mathcal{E}^\top X (X^\top X)^{-1} X^\top \mathcal{E}] = \sigma^2 \frac{\mathrm{trace}(X^\top X)}{NT} = \frac{\sigma^2(N+T+2)}{NT},
$$

which decays to zero as both $N$ and $T$ increase to infinity. The proof is hence completed. $\qquad \square$

## B.5  Proof of Theorem 1

To simplify the proof, we analyze a variant of our estimator computed via sample-splitting and cross-fitting. Specifically, we begin by dividing the entire dataset $\mathcal{D}$ into two subsets $\mathcal{D}^{(1)}$ and $\mathcal{D}^{(2)}$, each containing $N/2$ trajectories. Next, we separately apply the proposed two-way deconfounder algorithm detailed in Section 2 to the two data subsets to learn the transition functions and latent confounders. Denote them by $\widehat{\mathcal{P}}^{(j)}$, $\{\widehat{U}_i^{(j)}\}_i$ and $\{\widehat{W}_t^{(j)}\}_t$, respectively, for $j = 1, 2$. Finally, we construct two model-based estimators, given by

$$
\widehat{\eta}^{(1)} = \frac{2}{NT} \sum_{i \in \mathcal{D}^{(1)}} \sum_{t=1}^T \widehat{\mathbb{E}}^{(2)} \left( R_{i,t} \mid O_{i,1}, \widehat{U}_i^{(1)}, \left\{ \widehat{W}_{t'}^{(1)} \right\}_{t'=1}^t \right), \tag{3}
$$

and

$$
\widehat{\eta}^{(2)} = \frac{2}{NT} \sum_{i \in \mathcal{D}^{(2)}} \sum_{t=1}^T \widehat{\mathbb{E}}^{(1)} \left( R_{i,t} \mid O_{i,1}, \widehat{U}_i^{(2)}, \left\{ \widehat{W}_{t'}^{(2)} \right\}_{t'=1}^t \right), \tag{4}
$$

where $\widehat{\mathbb{E}}^{(j)}$ denotes the expectation that uses the transition function $\mathcal{P}^{(j)}$ to approximate $\mathbb{E}^\pi$, and the summation $\sum_{i \in \mathcal{D}^{(j)}}$ is carried over all trajectories in $\mathcal{D}^{(j)}$. We average them construct our final estimator $\widehat{\eta}^\pi = (\widehat{\eta}^{(1)} + \widehat{\eta}^{(2)})/2$.

Notice that in both (3) and (4), the transition function used to conduct the model-based estimator is independent of the initial observation and the estimated latent confounders. This avoids imposing additional VC-class type conditions on the transition network. We remark that sample splitting is widely used in statistics and machine learning [see e.g., Chernozhukov et al., 2018, Shi et al., 2021, Kallus and Uehara, 2022].

*Proof.* We focus on bounding $\mathbb{E}|\widehat{\eta}^{(1)} - \eta^\pi|$. The same upper bound applies to $\mathbb{E}|\widehat{\eta}^{(2)} - \eta^\pi|$ and $\mathbb{E}|\widehat{\eta}^\pi - \eta^\pi|$, thanks to the triangle inequality. To ease notation, we will remove the superscripts in $\widehat{\mathbb{E}}^{(2)}, \widehat{\mathcal{P}}^{(2)}, \widehat{U}_i^{(1)}, \widehat{W}_t^{(1)}$, and present them as $\widehat{\mathbb{E}}, \widehat{\mathcal{P}}, \widehat{U}_i, \widehat{W}_t$. Due to the use of cross-fitting, $\widehat{\mathbb{E}}$ and $\widehat{\mathcal{P}}$ are independent of $\widehat{U}_i$s and $\widehat{W}_t$s. The proof is divided into two steps. In the first step, we upper bound $\mathbb{E}|\widehat{\eta}^{(1)} - \overline{\eta}^{(1)}|$ where

$$
\overline{\eta}^{(1)} = \frac{2}{NT} \sum_{i \in \mathcal{D}^{(1)}} \sum_{t=1}^T \mathbb{E}^\pi (R_{i,t} | O_{i,1}, U_i, \{W_{t'}\}_{t'=1}^t).
$$

Next, in the second step, we upper bound $\mathbb{E}|\overline{\eta}^{(1)} - \eta^\pi|$. Finally, combining these two bounds leads to the upper bound for $\mathbb{E}|\widehat{\eta}^{(1)} - \eta^\pi|$.

**Step 1.** Recall that

$$\widehat{\eta}^{(1)} = \frac{2}{NT} \sum_{i \in \mathcal{D}^{(1)}} \sum_{t=1}^{T} \widehat{\mathbb{E}}(R_{i,t}|O_{i,1}, \widehat{U}_i, \{\widehat{W}_{t'}\}_{t'=1}^{t}).$$

Notice that the difference $\overline{\eta}^{(1)} - \widehat{\eta}^{(1)}$ can be decomposed into the sum of

$$\frac{2}{NT} \sum_{i \in \mathcal{D}^{(1)}} \sum_{t=1}^{T} \left[ \mathbb{E}^{\pi}(R_{i,t}|O_{i,1}, U_i, \{W_{t'}\}_{t'=1}^{t}) - \widehat{\mathbb{E}}(R_{i,t}|O_{i,1}, U_i, \{W_{t'}\}_{t'=1}^{t}) \right]$$

$$+ \quad \frac{2}{NT} \sum_{i \in \mathcal{D}^{(1)}} \sum_{t=1}^{T} \left[ \widehat{\mathbb{E}}^{\pi}(R_{i,t}|O_{i,1}, U_i, \{W_{t'}\}_{t'=1}^{t}) - \widehat{\mathbb{E}}(R_{i,t}|O_{i,1}, \widehat{U}_i, \{\widehat{W}_{t'}\}_{t'=1}^{t}) \right]$$

$$= \quad I_1 + I_2.$$

We first study $I_1$. Since $O_{i,1}$s and $U_i$s are independent of $W_t$s, the first line forms a sum of i.i.d. random variables. Thus, it converges to

$$I_1^* := \frac{1}{T} \sum_{t=1}^{T} \sum_{o,u} \left[ \mathbb{E}^{\pi}(R_{i,t}|O_{i,1} = o, U_i = u, \{W_{t'}\}_{t'=1}^{t}) - \widehat{\mathbb{E}}(R_{i,t}|O_{i,1} = o, U_i = u, \{W_{t'}\}_{t'=1}^{t}) \right]$$

$$\times \rho_0(o, u) = \frac{1}{T} \sum_{t=1}^{T} \sum_{o,u} I_{1,t}^*(o, u, \{W_{t'}\}_{t'=1}^{t}) \rho_0(o, u),$$

with the expected error $\mathbb{E}|I_1 - I_1^*|$ upper bounded by $cR_{\max}/\sqrt{N}$ for some constant $c > 0$ by Cauchy-Schwarz inequality.

It remains to study $I_1^*$, or equivalently, $I_{1,t}^*$ for each $t$. To begin with, consider the case where $t = 1$. It follows that

$$|I_{1,1}^*(o, u, W_1)| = \sum_a \pi(a|o)[R(a, o, u, W_1) - \widehat{R}(a, o, u, W_1)]$$

$$= \sum_{a,r,o'} \pi(a|o)r[\mathcal{P}(r, o'|a, o, u, W_1) - \widehat{\mathcal{P}}(r, o'|a, o, u, W_1)]$$

$$\leq R_{\max} \sum_a \pi(a|o)\text{TV}(\widehat{\mathcal{P}}(\bullet|a, o, u, W_1), \mathcal{P}(\bullet|a, o, u, W_1))$$

When $t = 2$, one can show that

$$|I_{1,2}^*(o_1, u, \{W_t\}_{t=1}^{2})| \leq R_{\max} \sum_{a_1} \pi(a_1|o_1)\text{TV}(\widehat{\mathcal{P}}(\bullet|a_1, o_1, u, W_1), \mathcal{P}(\bullet|a_1, o_1, u, W_1))$$

$$+ R_{\max} \sum_{a_2, o_2, a_1} \pi(a_2|o_2)\pi(a_1|o_1)\text{TV}(\widehat{\mathcal{P}}(\bullet|a_2, o_2, u, W_2), \mathcal{P}(\bullet|a_2, o_2, u, W_2))P(o_2|a_1, o_1, u, W_1).$$

Using the same argument, one can show that

$$|I_{1,t}^*(o_1, u, \{W_{t'}\}_{t'=1}^{t})| \leq R_{\max} \sum_{t'=1}^{t} \sum_{a_{t'}, o_{t'}, \cdots, a_1} \pi(a_t'|o_t') \prod_{j=1}^{t'-1} \left[ \pi(a_j|o_j)P(o_{j+1}|a_j, o_j, u, W_j) \right]$$

$$\times \text{TV}(\widehat{\mathcal{P}}(\bullet|a_{t'}, o_{t'}, u, W_{t'}), \mathcal{P}(\bullet|a_{t'}, o_{t'}, u, W_{t'})).$$

As such, we obtain

$$\sum_{t=1}^{T} \sum_{o,u} \mathbb{E}|I_{1,t}^*(o, u, \{W_{t'}\}_{t'=1}^{t})|\rho_0(o, u)$$

$$\leq \quad R_{\max} \sum_{t=1}^{T} \mathbb{E}^{\pi}[\text{TV}(\widehat{\mathcal{P}}(\bullet|A_{1,t}, O_{1,t}, U_1, W_t), \mathcal{P}(\bullet|A_{1,t}, O_{1,t}, U_1, W_t))].$$

Using the change of measure theorem, the right-hand-side can be upper bounded by

$$CTR_{\max}\mathbb{E}[\text{TV}(\widehat{\mathcal{P}}(\bullet|A_{1,t}, O_{1,t}, U_1, W_t), \mathcal{P}(\bullet|A_{1,t}, O_{1,t}, U_1, W_t))],$$

where $C$ denotes the single-policy concentration coefficient defined in Assumption 1. By Cauchy-Schwarz inequality, the above expression can be further bounded from above by $CTR_{\max}\varepsilon_{\mathcal{P}}$.

Using similar arguments, we can upper bound $|I_2|$ by

$$\frac{R_{\max}}{N} \sum_{i=1}^{N} \sum_{t=1}^{T} \max_{a,o}[\text{TV}(\widehat{\mathcal{P}}(\bullet|a, o, \widehat{U}_i, \widehat{W}_t), \widehat{\mathcal{P}}(\bullet|a, o, U_i, W_t))].$$

As neural networks are Lipschitz continuous functions of their parameters, the above total variation norm is proportional to $\|(\widehat{U}_i^\top, \widehat{W}_t^\top)^\top - (U_i^\top, W_t^\top)\|_2$. Consequently, under Assumption 4, the above expression is upper bounded by $cTR_{\max}\varepsilon_{U,W}$ for some constant $c > 0$. This completes the proof of Step 1.

**Step 2.** According to the DGP described in Section 2, $(U_1, O_{1,1}), \cdots, (U_n, O_{n,1})$ are i.i.d., $\eta^\pi$, and are independent of $\{W_t\}_t$. $\eta^\pi$ can thus be represented by $T^{-1} \sum_{t=1}^{T} \mathbb{E}^\pi(R_{1,t})$. We further decompose $|\bar{\eta}^{(1)} - \eta^\pi|$ into the following two components:

$$|\bar{\eta}^{(1)} - \eta^\pi| = \left| \frac{2}{NT} \sum_{i \in \mathcal{D}^{(1)}} \sum_{t=1}^{T} \mathbb{E}^\pi(R_{i,t}|O_{i,1}, U_i, \{W_{t'}\}_{t'=1}^t) - \frac{1}{T} \sum_{t=1}^{T} \mathbb{E}^\pi(R_{1,t}) \right|$$

$$\leq \left| \frac{2}{NT} \sum_{i \in \mathcal{D}^{(1)}} \sum_{t=1}^{T} \mathbb{E}^\pi(R_{i,t}|O_{i,1}, U_i, \{W_{t'}\}_{t'=1}^t) - \frac{1}{T} \sum_{t=1}^{T} \mathbb{E}^\pi(R_{1,t}|\{W_{t'}\}_{t'=1}^t) \right|$$

$$+ \left| \frac{1}{T} \sum_{t=1}^{T} \mathbb{E}^\pi(R_{1,t}|\{W_{t'}\}_{t'=1}^t) - \frac{1}{T} \sum_{t=1}^{T} \mathbb{E}^\pi(R_{1,t}) \right| = I_3 + I_4.$$

Conditional on $\{W_t\}_t$, $I_3$ corresponds to a sum of i.i.d. random variables. An application of Cauchy-Schwarz inequality implies that $\mathbb{E}I_3 \leq \sqrt{\mathbb{E}I_3^2} \leq cR_{\max}/\sqrt{N}$, for some constant $c > 0$. Similarly, under Assumption 5, we have

$$\mathbb{E}I_4 \leq \sqrt{\mathbb{E}I_4^2} \leq \frac{R_{\max}}{T} \sqrt{\sum_{1 \leq t_1, t_2 \leq T} \rho(t_1, t_2)} = O(R_{\max}T^{\alpha-1}).$$

The proof is hence completed by combining the results in both steps. $\qquad\square$

## C Implementation.

### C.1 Details for loss function

The final loss is defined as follows:

$$L(\mathcal{D}; \{u_i\}_i, \{w_t\}_t) = (1 - \alpha) \cdot \frac{1}{2} \sum_{i,t} \left\{ [\widehat{\mu} - \varphi_{i,t}]^\top \widehat{\Sigma}^{-1} [\widehat{\mu} - \varphi_{i,t}] + \log \det \widehat{\Sigma} \right\}$$

$$+ \alpha \cdot \sum_{i,t} \text{CrossEntropy}(a_{i,t}, \widehat{\pi}_b(a_{i,t}|o_{i,t}, u_i, w_t)),$$

where loss weighting $\alpha \in (0, 1)$ is a hyperparameter, $\varphi_{i,t} = \left(r_{i,t}, o_{i,t+1}^\top\right)^\top$, and $(a_{i,t}, o_{i,t}, r_{i,t}, o_{i,t+1})$ is sampled from the offline dataset $\mathcal{D}$, $\widehat{\Sigma} = \text{diag}(\widehat{\sigma})$, $\det \widehat{\Sigma}$ represents the determinant of $\widehat{\Sigma}$, and CrossEntropy is the cross-entropy loss.

### C.2 Implementation Details for Two-way Deconfounder Model.

The Two-way Deconfounder Model described in Section 3 was implemented in Pytorch and trained on an NVIDIA GeForce RTX 3090. For every task, we repeat it 20 times and the results are aggregated over 20 runs. The training time of 4 to 12 hours hours (depending on the task) on a single NVIDIA GeForce RTX 3090. Each dataset undergoes a 75/25 split for training, validation respectively. Using the validation set, we perform hyperparameter optimization using many iterations of grid

search to find the optimal values for hyperparameters. The search range for each hyperparameter is described as follows, learning rate $lr \in [0.005, 0.001]$, Batch size $bs \in [2^8, 2^9, 2^{10}, 2^{11}, 2^{12}]$, weight decay $\lambda \in [0.01, 0.0001]$, two-way embedding dimension $d_{tw} \in [2^1, 2^2, 2^3]$, Loss weighting $\alpha \in [0.0, 0.3, 0.5, 0.7]$.

### C.3 Implementation Details for Ablation Study.

Variants of TWD are model-based methods, model-based method with two-way embedding(TWD-TO,TWD-MLP) and model-based method with one-way embedding(TWD-NI,TWD-NT)Yu et al. [2020], Kausik et al. [2022], where the first can be applied to the two-way setting , and the second is originally designed for the one-way case. We adopt the same embedding approach for variants of TWD as our method. TWD-TO can be seen as a special case of our method, which only remove the CrossEntropy loss of the behavior policy. TWD-MLP remove the neural tensor network. However, the biggest difference between TWD-NI or TWD-NT and our method is that it use the one-way embedding based on Kausik et al. [2022] removing the embedding of trajectory-invariant or time-invariant unmeasured confounders. We provide Change details of those variants compared to TWD in the following.

- TWD-TO: Because it removes the CrossEntropy loss of the behavior policy, the loss function is as follows,

$$L\left(\mathcal{D}; \{u_i\}_i, \{w_t\}_t\right) = \sum_{i,t} [\{\left[\widehat{\mu} - \varphi_{i,t}\right]^\top \widehat{\Sigma}^{-1} \left[\widehat{\mu} - \varphi_{i,t}\right] + \log \det \widehat{\Sigma}\}],$$

  other model settings are the same as TWD.

- TWD-MLP: Its conditional mean and variance functions are modeled jointly with the behavior policy,

$$(\widehat{\mu}^\top, \widehat{\sigma}^\top)^\top = \mathrm{MLP}_{\mathcal{P}}(a_{i,t}, \mathrm{MLP}(o, u_i, w_t)),$$
$$\widehat{\pi}_b(\bullet \mid o, u_i, w_t) = \mathrm{MLP}_{\pi_b}(\mathrm{MLP}(o, u_i, w_t)),$$

  It replaces the neural tensor network with a simple MLP and MLP is the multilayer perceptron that take $o$, $u_i$ and $w_t$ as input.

- TWD-NI: Its conditional mean and variance functions are modeled jointly with the behavior policy,

$$(\widehat{\mu}^\top, \widehat{\sigma}^\top)^\top = \mathrm{MLP}_{\mathcal{P}}(a_{i,t}, \mathrm{NI}(o, w_t)),$$
$$\widehat{\pi}_b(\bullet \mid o, w_t) = \mathrm{MLP}_{\pi_b}(\mathrm{NI}(o, w_t)),$$

  It removes the individual embedding and NI is the multilayer perceptron that take $o$ and $w_t$ as input.

- TWD-NT: Its conditional mean and variance functions are modeled jointly with the behavior policy,

$$(\widehat{\mu}^\top, \widehat{\sigma}^\top)^\top = \mathrm{MLP}_{\mathcal{P}}(a_{i,t}, \mathrm{NT}(o, u_i)),$$
$$\widehat{\pi}_b(\bullet \mid o, u_i) = \mathrm{MLP}_{\pi_b}(\mathrm{NT}(o, u_i)),$$

  It removes the time embedding and NT is the multilayer perceptron that take $o$ and $u_i$ as input.

## D  Simulation details

### D.1  A linear simulation setting

Let $\mathcal{O} = \mathbb{R}, \mathcal{U} = \mathbb{R}, \mathcal{W} = \mathbb{R}, \mathcal{R} = \mathbb{R}$, and $\mathcal{A} = \{1, 0\}$. For evey $i$ and $t$, $u_i$ and $w_t$ are sampled a Normal distribution $\mathcal{N}(0, 1)$. The observed data is generated as follows,

**The transition model.** For each $i$ and $t$, given $(O_{i,t}, A_{i,t}, U_i, W_t)$, We generate

$$O_{i,t+1} = 0.7 O_{i,t} + A_{i,t} + 2U_i + 2W_t - 0.5 + e_s$$

where the random error $e_s \sim \mathcal{N}(0, 1)$

**The reward model.** For each $i$ and $t$, given $(O_{i,t}, A_{i,t}, U_i, W_t)$, We generate

$$R_{i,t} = O_{i,t} + 3A_{i,t} + 2U_i + 2W_t + e_r$$

where the random error $e_r \sim \mathcal{N}(0, 2)$

**Behavior policy.** For each $i$ and $t$, given $(O_{i,t}, U_i, W_t)$, We generate

$$p(A_{i,t} = 1 \mid O_{i,t}, U_i, W_t) = \text{sigmoid}(O_{i,t} + U_i + W_t)$$

**Target policy.** $p(A_{i,t} = 1) = 0.5$

We generate datasets respectively under the behavioral and target policy,. Under the assumptions of one-way unmeasured confounders, two-way unmeasured confounders, and unconstrained unmeasured confounders, respectively, we use the least squares method to fit the model parameters. Then, we use the learned model to perform the off-policy evaluation. The number of trajectory is $N = \{200, 300, 400, 500, 600, 700, 800\}$. The length of the horizon is fixed at 50. we calculate MSE error in fitting the observed data and MSE error in off-policy evaluation, respectively. Two-way unmeasured confounders outperformed the other two on off-policy evaluation, as shown in the Figure 1.

## D.2 Simulated Dynamic Process

We first describe the detailed setting for the simulated data. Let $\mathcal{O} = \mathbb{R}^4, \mathcal{U} = \mathbb{R}, \mathcal{W} = \mathbb{R}, \mathcal{R} = \mathbb{R}$, and $\mathcal{A} = \{1, 0\}$. We let $T = 50$ and the number of data trajectories $N = \{250, 500, 1000, 1500, 2000\}$. We consider a four-dimensional variable $O_{i,t} = (O_{i,t}^1, O_{i,t}^2, O_{i,t}^3, O_{i,t}^4)$ whose initial distribution is given by $\mathcal{N}(\mathbf{0}_4, \mathbf{I}_4)$ where $\mathbf{I}_4$ denotes a four-dimensional identity matrix. The individual-invariant unmeasured confounders $\{U_i\}_i$ are sampled a Normal distribution $\mathcal{N}(0, 1)$. The time-invariant unmeasured confounders $\{W_t\}_t$ are sampled a Normal distribution $\mathcal{N}(0, 1)$. The data generating process is outlined as follows, reward function and transition function are given by $R_{i,t} = f_r(O_{i,t}, A_{i,t}, U_i, W_t, \Gamma) + e_r$, $e_r \sim \mathcal{N}(0, 1)$, and $O_{i,t+1} = f_o(O_{i,t}, A_{i,t}, U_i, W_t, \Gamma) + e_o$, $e_o \sim \mathcal{N}(\mathbf{0}_4, \mathbf{I}_4)$, the behavior policy satisfies $\mathbb{P}(A_{i,t} = 1 \mid O_{i,t}, U_i, W_t) = \text{sigmoid}(f_a(O_{i,t}, U_i, W_t, \Gamma))$. $\Gamma$ is the confounding strength parameter. We set $\Gamma = 1$ in this section. $f_o(\cdot)$, $f_r(\cdot)$ and $f_a(\cdot)$ are the functions of input variable on the next observation, immediate reward, and action, respectively.

**The transition model.** For each $i$ and $t$, given $(O_{i,t}, A_{i,t}, U_i, W_t)$ and $\Gamma$, We generate

$$O_{i,t+1} = \mu_s O_{i,t} + \Gamma \cdot \beta_o f(U_i, W_t) + \lambda_o A_{i,t} \mathbf{I}_4 + b_o + e_o$$

where $\mathbf{I}_4 = [1, 1, 1, 1]^\top$, the random error $e_o \sim \mathcal{N}(\mathbf{0}_4, \mathbf{I}_4)$ with $\mathbf{I}_4$ denoting the 4-by-4 identity matrix,

- $\mu_o = \begin{bmatrix} 0.8 & 0 & 0 & 0 \\ 0 & 0.8 & 0 & 0 \\ 0 & 0 & 0.8 & 0 \\ 0 & 0 & 0 & 0.8 \end{bmatrix}$,

- $\beta_o = \begin{bmatrix} 0.1 & 0 & 0 & 0 \\ 0 & 0.1 & 0 & 0 \\ 0 & 0 & 0.1 & 0 \\ 0 & 0 & 0 & 0.1 \end{bmatrix}$,

- $\lambda_o = [1, 1, 1, 1]^\top$,

- $b_o = [-0.5, -0.5, -0.5, -0.5]^\top$,

- $f_o(U_i, W_t) = [u_i - w_t, u_i + w_t, -u_i - w_t, -u_i + w_t]^\top$.

**The reward model.** For each $i$ and $t$, given $(O_{i,t}, A_{i,t}, U_i, W_t)$ and $\Gamma$, We generate

$$R_{i,t} = \mu_r O_{i,t} + \Gamma \cdot \beta_r f(U_i, W_t) + \lambda_r A_{i,t} + e_r$$

where the random error $e_r \sim \mathcal{N}(0, 1)$, $\beta_r = 3.0$, $\lambda_r = 2.5$,

- $\mu_r = [0.25, 0.25, 0.25, 0.25]^\top$,

- $f_r\left(U_i, W_t\right) = u_i \cdot w_t.$

**Behavior policy.** For each $i$ and $t$, given $(O_{i,t}, U_i, W_t)$ and $\Gamma$, We generate

$$p\left(A_{i,t} = 1 \mid O_{i,t}, U_i, W_t\right) = \text{sigmoid}\left(\mu_a O_{i,t} - 4 + \Gamma \cdot \beta_a f_a\left(U_i, W_t\right)\right)$$

where $\beta_a = 3$,

- $\mu_a = [0.25, 0.25, 0.25, 0.25]^\top$,
- $f_r\left(U_i, W_t\right) = u_i \cdot w_t + u_i + w_t.$

**Target policy.**

- Target policy A: $p(A_{i,t} = 1) = 0.3$.
- Target policy B: $p(A_{i,t} = 1) = 0.5$.

### D.3 Tumor Growth simulated data setup

In this section, we give brief description about Tumor Growth environment setting and experiment details. We set up many subsets randomly with different numbers of patients who received treatments sequentially over 60 stages. To be specific, we randomized patients into three groups, with each patient having a group label $S_i \in \{1, 2, 3\}$.

**The transition model**. We apply the unobserved confounding variables to the PK-PD model:

- $V(t) = (1 + \rho \log\left(\frac{K}{V(t-1)}\right) - \beta_c C(t) - \left(\alpha A^r(t) + \beta A^r(t)^2\right) + e_t) V(t-1)$. where $\rho$ and $K$ are model parameters sampled for each patient according to prior distributions in Geng et al. [2017], Lim [2018], Bica et al. [2020] and $e_t \sim N\left(0, 0.01^2\right)$. Specifically, $\beta_c$, $\alpha$ and $\beta$ are model parameters sampled for representing specific characteristics which affect with patient's response to chemotherapy and radiotherapy according to randomly subclassing patients into 3 different groups $S_i \in \{1, 2, 3\}$ in Geng et al. [2017], Lim [2018], Bica et al. [2020], which are influenced by genetic factors [Bartsch et al., 2007] and can be regarded as time-invariant confounders.
- $C(t) = C(t-1)/2 + 5 \cdot A^c(t)$, which relies on the chemo treatment action and exponentially decays over time.

**The reward model**. The reward model consists of three parts:

- *pathological-health reward*: A negative reward $R_n$ for penalising the patient if tumor size is large or accepts too much drugs and radio: $1.5 \exp(-V(t))$.
- *side-effect reward*: A positive reward $R_p$ for accepting treatments when take action at time step $t$: $\exp(-(A^r(t) + A^c(t))^2)$
- *reward imposed by unmeasured confounders*: A reward $R_{TW}$ about the two-way confounder parameter: $(4 \cdot sigmoid(S_i - 2) - \sin(0.1\pi t))$.
- *total reward*: $R = R_n + R_P + R_{TW} + e_r, e_r \sim \mathcal{N}\left(0, 1\right)$

**Behavior policy.** We introduce two-way unmeasured confounders to treatment assignment policy. For each $i$ and $t$, We generate,

$$A^c(t) \sim Bernoulli(\text{sigmoid}(\frac{\gamma_c}{D_{\max}}\left(D(t) - \theta_c\right) + (3 \cdot sigmoid(S_i - 2) - 0.75 \cdot \sin(0.1\pi t))))$$

$$A^r(t) \sim Bernoulli(\text{sigmoid}(\frac{\gamma_r}{D_{\max}}\left(D(t) - \theta_r\right) + (3 \cdot sigmoid(S_i - 2) - 0.75 \cdot \sin(0.1\pi t))))$$

where $D(t)$ is tumor diameters, $D_{\max}$ is the half maximum size of the tumor, $\theta_c$ and $\theta_r$ are fixed such that $\theta_c = \theta_r = D_{\max}/2$ and $\gamma_c$ and $\gamma_r$ are also fixed such that $\gamma_c = \gamma_r = 10.0$ .

**Target policy.**

- Target policy A: $A^c(t), A^r(t) \sim Bernoulli(0.05)$.

- Target policy B: $A^c(t), A^r(t) \sim Bernoulli(p)$, where $p$ is related to the conditions of patients:

$$p = \begin{cases} 0.2, & if\ Y_t > 88, \\ 0.1, & if\ 44 < Y_t \leq 88, \\ 0.05, & if\ 5 < Y_t \leq 44, \\ 0.01, & otherwize \end{cases} \tag{5}$$

## D.4 A real-world example: MIMIC-III.

We show how the Two-way Deconfounder can be applied to a real-world medical setting using the Medical Information Mart for Intensive Care (MIMIC-III) database [Johnson et al., 2016]. We extract 3,707 patients with trajectories up to 20 timesteps. Following the analysis of Zhou et al. [2023b], we define a $5 \times 5$ action space by discretizing both vasopressors fluid $A^v$ and intravenous $A^i$ fluid interventions into 5 levels. We define the reward $R_{i,t} = SOFA_{i,t} - SOFA_{i,t+1}$ as the difference between current SOFA score and next SOFA score, so a lower reward indicates a higher risk of mortality. We extract 12 patient covariates as observation space, including Glasgow coma scale,Systolic blood pressure, Weight, SOFA, SIRS, Fraction of inspired oxygen, Blood Urea Nitrogen, Creatinine, Serum Glutamic-Oxaloacetic Transaminase, Partial Pressure of Oxygen , Total Bilirubin, Platelet Count. Considering the complexity of the real data, we incorporate the normalization step before modeling them.

**Target policy.**

- the randomized policy: $P(A^v = 0) = P(A^v = 1) = p(A^v = 2) = P(A^v = 3) = p(A^v = 4) = 0.2, P(A^i = 0) = P(A^i = 1) = p(A^i = 2) = P(A^i = 3) = p(A^i = 4) = 0.2$.
- the high dose polic: $P(A^v = 3) = p(A^v = 4) = 0.5, P(A^i = 3) = p(A^i = 4) = 0.5$.
- the low dose polic: $P(A^v = 0) = 0.4, P(A^v = 1) = p(A^v = 2) = 0.3, P(A^i = 0) = 0.4, P(A^i = 1) = p(A^i = 2) = 0.3$.
- the tailored policy:
  - if normalized SOFA score $\geq 0.95$, $p(A^v = 4) = p(A^v = 3) = 0.3, p(A^v = 2) = 0.2$ and $P(A^v = 0) = P(A^v = 1) = 0; p(A^i = 4) = p(A^i = 3) = 0.3, p(A^i = 2) = 0.2$ and $P(A^i = 0) = P(A^i = 1) = 0$.
  - if $0.95 >$ normalized SOFA score $\geq 0.7$, $p(A^v = 4) = p(A^v = 3) = 0.1, p(A^v = 2) = 0.2, p(A^v = 1) = p(A^v = 0) = 0.3; p(A^i = 4) = p(A^i = 3) = 0.1, p(A^i = 2) = 0.2, p(A^i = 1) = p(A^i = 0) = 0.3$.
  - if normalized SOFA score $< 0.7$, $p(A^v = 0) = 0.8, p(A^v = 1) = 0.2 and p(A^v = 2) = p(A^v = 1) = p(A^v = 0) = 0; p(A^i = 0) = 0.8, p(A^i = 1) = 0.2 and p(A^i = 2) = p(A^i = 1) = p(A^i = 0) = 0$.

## D.5 Sensitivity analysis

In two simulated datasets, the modified reward model and behavior policy according to sensitivity parameter $\Gamma$ are as follows,

**Simulated Dynamic Process.**

- The reward model: For each $i$ and $t$, given $(O_{i,t}, A_{i,t}, U_i, W_t)$ and $\Gamma$, We generate

$$R_{t,t} = \mu_r O_{t,t} + \Gamma \cdot \beta_r f(U_i, W_t) + \lambda_r A_{i,t} + e_r$$

  where the random error $e_r \sim \mathcal{N}(0, 1)$, $\beta_r = 3.0$, $\lambda_r = 2.5$,
  - $\mu_r = [0.25, 0.25, 0.25, 0.25]^\top$,
  - $f_r(U_i, W_t) = u_i \cdot w_t$.
- Behavior policy: For each $i$ and $t$, given $(O_{i,t}, U_i, W_t)$ and $\Gamma$, We generate

$$p(A_{i,t} = 1 \mid O_{i,t}, U_i, W_t) = \text{sigmoid}(\mu_a O_{i,t} - 4 + \Gamma \cdot \beta_a f_a(U_i, W_t))$$

  where $\beta_a = 3$,

- $\mu_a = [0.25, 0.25, 0.25, 0.25]^\top$,
- $f_r(U_i, W_t) = u_i \cdot w_t + u_i + w_t$.

**Tumor Growth.**

- The reward model: $R = R_n + R_P + R_{TW} + e_r$, where $e_r \sim \mathcal{N}(0,1)$ and $R_{TW} = \Gamma \cdot (4 \cdot sigmoid(S_i - 2) - \sin(0.1\pi t))$
- Behavior policy: For each $i$ and $t$, We generate,

$$A^c(t) \sim Bernoulli(\text{sigmoid}(\frac{\gamma_c}{D_{\max}}(D(t) - \theta_c)$$
$$+ \Gamma \cdot (3 \cdot sigmoid(S_i - 2) - 0.75 \cdot \sin(0.1\pi t))))$$
$$A^r(t) \sim Bernoulli(\text{sigmoid}(\frac{\gamma_r}{D_{\max}}(D(t) - \theta_r)$$
$$\Gamma \cdot (3 \cdot sigmoid(S_i - 2) - 0.75 \cdot \sin(0.1\pi t))))$$

where $D(t)$ is tumor diameters, $D_{\max}$ is the half maximum size of the tumor, $\theta_c$ and $\theta_r$ are fixed such that $\theta_c = \theta_r = D_{\max}/2$ and $\gamma_c$ and $\gamma_r$ are also fixed such that $\gamma_c = \gamma_r = 10.0$ .

