# OpenReview forum: "Two-way Deconfounder for Off-policy Evaluation in Causal Reinforcement Learning"
_NeurIPS.cc/2024/Conference — NeurIPS 2024 poster_

### Official Review · Reviewer_P4Ev · 2024-07-09

**Soundness:** 3
**Presentation:** 2
**Contribution:** 2
**Rating:** 4
**Confidence:** 3

**Summary:**

This paper considers a setting in which there are observed tuples $(O_{i,t}, A_{i, t}, R_{i, t})$ with $ 1\leq i \leq N$, $1 \leq t \leq T$;
Here, $O_{i,t}$ represents observable quantities, $A_{i,t}$ is an action generated by a behavioural policy, and $R_{i,t}$ is the associated reward.
The goal of this paper is off-policy evaluation; specifically, the authors aim to estimate the expected cumulative reward under a given policy that differs from the behavioural policy.

The authors propose a generative model of the observed trajectories. The specific contribution of this work is to consider a two-way confounder, which as I understand a tuple of latent variables $(U_i, W_t)$ representing subject-and time-varying components, respectively.

**Strengths:**

The proposed two-way confounding model can be considered original. The authors considers separating a confounder into a time- and subject-invariant parts, and model their interaction by a neural network. This may be considered as a low-rank approximation (analogously to nonlinear matrix-factorisation), and reduces the number of the latent variables from $NT$ to $N + T$.

The empirical evaluation shows that the proposed method performs well. The authors also experimentally demonstrates the sensitivity to the two-way confounder assumption.

**Weaknesses:**

# Clarity

## Major issues:
1. Line 104: It is unclear what $\mathbb{E}^\pi$ denotes. If the expectation is simply taken with respect to the dynamics generated by $\pi$, doesn't this have an effect from the confounders?
2. Introducing the estimator (1) before the model would help the reader to understand the quantities to be estimated and hence the model.
3. The assumption on the data generating process is not straightforward to understand. In the unconstrained unmeasured confounding (UUC) setting, the trajectories cannot be independent, and the data assumption (Line 89) alone  already excludes the UUC setting. At the same time, for the proposed TWUC setting, the presence of the subject-invariant latent $W_t$ renders the trajectories dependent on each other. Using a graphical model would help clarify the assumption.

## Minor issues:
1. The $\cup$ notation is undefined.
2. Line 187: Gaussian Mixture Model. The transition kernel does not look like a mixture. Why the term?
3. L 287: forr

# Theoretical justification
While I realise that the focus of the paper is on the empirical performance, the supporting theory is relatively weak.

1. The validity of the estimator (1) is claimed, but there is not reference to this claim. Moreover, the expectation needs to be estimated, and calling (1) an estimator sound odd to me.
2. I have not been able to read though the proof, but the first paragraph already casts some doubt. Why does estimand $\eta$ depend on the latent and why are they not marginalised? (this might explain what I don't understand about the assumption). It seems that the only variability comes from the initial observation rather than the whole trajectories (since the confounders are fixed). The consistency result therefore has limited reliability,

**Questions:**

NA

**Limitations:**

Yes

---

> ### Author Rebuttal · Authors · 2024-08-06
>
> - __Clarification of the DGP.__\
> We agree with your points and will replace "independent trajectories" with "trajectories". "Independent" was used to indicate conditional independence between trajectories given latent two-way fixed effects, but this was confusing. Following your suggestion, we use different use different graphical models to clarify DGP under different assumptions in Figure 1 of the PDF file. We will include this figure if our paper is accepted.
> - __$\eta^\pi$'s dependence on latent variables and proof sketch.__\
> To validate our theory and save your time, we provide an outline of our proof below. We first clarify $\eta^\pi$'s dependence on latent variables:
>   1. We assume two-way unmeasured confounders are fixed, aligning with existing literature on two-way fixed effects regression.
>   2. While latent factors $U_i$s and $W_t$s are considered fixed, not all observations $O_{i,t}$s are fixed. Thus, **the variability of $R_{i,t}$ comes from the initial observation and all subsequent intermediate observations along the trajectory**, generated randomly by the transition function.
>   3. The estimation error of $\eta^\pi$ arises from variability in the initial observation, estimated transition dynamics, latent factors, and distributional shift between behavior and target policy, as highlighted in the first two terms of our error bound in Theorem 1.
>
>   Meanwhile, our theory can also accommodate random two-way unmeasured confounders by bounding the deviation between $(NT)^{-1}\sum_{i,t} \mathbb{E}^\pi[R_{i,t}|U_i,\\{W_{t'}\\}\_{t'=1}^t]$ and $T^{-1}\sum_t \mathbb{E}^\pi[R_{1,t}]$. This deviation can be decomposed into two terms:
>   $$\Big[\frac{1}{NT}\sum_{i,t} \mathbb{E}^\pi[R_{i,t}|U_i,\\{W_{t'}\\}\_{t'=1}^t]-\frac{1}{T}\sum_t \mathbb{E}^\pi[R_{1,t}|\\{W_{t'}\\}\_{t'=1}^t]\Big]+\Big[\frac{1}{T}\sum_t\mathbb{E}^\pi[R_{1,t}|\\{W_{t'}\\}\_{t'=1}^t]-\frac{1}{T}\sum_t\mathbb{E}^\pi[R_{1,t}]\Big].$$
>   Assuming $U_i$s are i.i.d. and independent from $W_t$s, we apply Hoeffding's inequality to the first term, resulting in an order of $R_{max}\sqrt{2N^{-1}\log (2/\delta)}$ with probability at least $1-\delta$. The second term requires assuming that the time series exhibits certain mixing properties (e.g., $\beta$-mixing), allowing us to treat each $R_{1,t}$ as dependent only on recent $W_{t'}$s. Using Berbee’s coupling lemma (Berbee 1987) and approximating this term with sums of i.i.d. random variables, we apply Hoeffding's inequality again to establish the tail inequality for the second term. We are happy to revise our theory and proof to incorporate these changes if our paper is accpeted.
>
>   Finally, to facilitate understanding of Theorem 1, we provide a _proof sketch_. We decompose $|\hat{\eta}^\pi - \eta^\pi|$ into two terms, $I_1$ and $I_2$. $I_2$ represents the absolute difference between the mean of the value function and its expectation, given the latent confounders and transition function are true values. The only difference between the two terms is the expectation over the initial state. Based on the boundedness of the value function (Assumption 2) and the conditional independence of each trajectory given all latent confounders, we can apply Hoeffding's inequality. This gives $I_2$ an order of $R_{max}\sqrt{2N^{-1}\log (2/\delta)}$ with probability at least $1-\delta$. $I_1$ represents the absolute difference between the estimator $\hat{\eta}^{\pi}$ (the mean of the estimated value function) and the mean of the true value function. Using the Bellman equation, we transform the difference into a function of differences between the estimated and true values of the transition function, i.e., a function of $\operatorname{TV}(\hat{\mathcal{P}}(\bullet|a_t,o_t,\hat{u}\_i,\hat{w}\_t)-\mathcal{P}(\bullet|a_t,o_t,u_i,w_t))$. Furthermore, the function of $\operatorname{TV}(\hat{\mathcal{P}}(\bullet|a_t,o_t,\hat{u}\_i,\hat{w}\_t)-\mathcal{P}(\bullet|a_t,o_t,u_i,w_t))$ can be decomposed into the sum of the functions of $\operatorname{TV}(\hat{\mathcal{P}}(\bullet|a_t,o_t,\hat{u}\_i,\hat{w}\_t)-\hat{\mathcal{P}}(\bullet|a_t,o_t,u_i,w_t))$ (denoted as $I_3$) and $\operatorname{TV}(\hat{\mathcal{P}}(\bullet|a_t,o_t,u_i,w_t)-\mathcal{P}(\bullet|a_t,o_t,u_i,w_t))$ (denoted as $I_4$). Based on Assumption 4 and the Lipschitz continuity of the neural network, we can determine that the order of $I_3$ is $\varepsilon_{U,W,\delta}$ with probability at least $1-\delta$. Finally, by leveraging Assumptions 1 and 3, we find that the order of $I_4$ is $\varepsilon_{\mathcal{P},\delta}$ with probability at least $1-\delta$. Combining this with all previously derived conclusions, we can then establish the order of $|\hat{\eta}^\pi-\eta^\pi|$ with probability at least $1-3\delta$.
>
>
>
> - __Equation (1).__
>   - __Is (1) an estimator?__: As you commented, (1) is not our final estimator, since the expectation needs to be approximated based on the Monte Carlo method detailed on Page 5, Lines 205 to 210. It serves as an **intermediate estimator** that is unbiased to the evaluation target (see our justification of the unbiasedness in the next response). We will make this clear to avoid potential confusion.
>   - __Validity of (1)__: We claimed Equation (1) is unbiased to $\eta^\pi$ in Line 200. Demonstrating this unbiasedness is straightforward: Recall that $\eta^\pi$ is the average expected reward across $N$ individuals over $T$ time points. Applying the law of total expectation, it can be expressed as $(NT)^{-1}\sum_{i,t}\mathbb{E}^\pi[R_{i,t}|O_{i,1},U_i,\\{W_{t'}\\}\_{t'=1}^t]$, which equals Equation (1).
>   - __Introducing (1) earlier__: Following your suggestion, we will introduce (1) before the model if our paper is accepted.
> - __Minor issues.__\
> We apologize for the typos. In line 126, we will revise $Z_{i,t}=U_i\cup W_t$ to $Z_{i,t}=(U_i^\top,W_t^\top)^\top$, to avoid using the undefined notation $\cup$. Additionally, we will replace 'Gaussian Mixture Model' with 'Gaussian Model' in line 187.

---

> > ### Comment · Reviewer_P4Ev · 2024-08-12
> >
> > I thank the authors for the clarifications. As a follow-up comment, I think it is confusing to analyse the case where the latent variables are fixed given that the objective does not contain any latent (as in the general response), and this should be corrected (as proposed).
> > The theoretical analysis currently does not seem to be well linked to the proposed training method, it remains unclear when the four assumptions can be achieved.
> > For this reason, I still find it difficult to provide a strong support for acceptance, while I appreciate the empirical nature (and performance) of the present work.

---

> > > ### Author Response · Authors · 2024-08-14
> > >
> > > We sincerely thank the referee for the positive feedback on the empirical nature and performance of our paper. Below, we address the remaining comments on our theoretical analysis.
> > >
> > > - __Latent v.s. fixed variables.__
> > > We will revise this as suggested. As outlined in the rebuttal, the error bound remains similar to Theorem 1, with an additional term of $c_2 R_\max (N^{-1/2}+T^{-1/2})$ up to some logarithmic factors, to account for the randomness of the latent factors. We hope this clarification resolves your concerns.
> > >
> > > - __Clarification on the assumptions.__
> > > This issue was not raised in the official review, but we believe Assumptions 1–4 are mild and achievable. We hope our response addresses your concerns.
> > >
> > >    * Assumptions 1 and 2 are standard in the reinforcement learning literature (e.g., [1], [2], [3], [4], [11], [12], [13], [14]). Though space limits citations, these assumptions are widely applied in offline policy optimization and off-policy evaluation.
> > >    * Assumption 3 is concerned with the estimation error of the transition function. This assumption is flexible, as $\varepsilon_{\mathcal{P},\delta}$ can be adjusted to a larger value to meet the condition. In our implementation, we use a conditional Gaussian model for the transition function, following [6]. The total variation bound, measured by $\varepsilon_{\mathcal{P},\delta}$, thus reduces to the estimation errors of the mean and covariance functions. Using neural networks for function approximation allows the estimator to achieve an optimal non-parametric convergence rate (e.g., [7], [8], [9]). This yields the specific form of $\varepsilon_{\mathcal{P},\delta}$. Even without the conditional Gaussian model, deep generative learning algorithms with theoretical guarantees can be employed, with error bounds provided in studies like [10].
> > >    * Assumption 4 is concerned with the estimation error of the latent confounders. Like Assumption 3, it is flexible, as $\varepsilon_{U,W,\delta}$ can be adjusted to a larger value. Additionally, under a two-way additive model assumption, the factors $\\{U\_i\\}\_i$ and $\\{W\_t\\}\_t$ can be estimated at orders of $\sqrt{T^{-1/2}\log (N/\delta)}$ and $\sqrt{N^{-1/2}\log (T/\delta)}$ with probabilities at least $1-O(\delta)$ (see, [15]). This provides the detailed forms of $\varepsilon_{U,W,\delta}$.
> > >
> > > [1] Chen J, Jiang N. Information-theoretic considerations in batch reinforcement learning[C]//International Conference on Machine Learning. PMLR, 2019: 1042-1051. \
> > > [2] Fan J, Wang Z, Xie Y, et al. A theoretical analysis of deep Q-learning[C]//Learning for dynamics and control. PMLR, 2020: 486-489.\
> > > [3] Liu Y, Swaminathan A, Agarwal A, et al. Provably good batch off-policy reinforcement learning without great exploration[J]. Advances in neural information processing systems, 2020, 33: 1264-1274. \
> > > [4] Uehara M, Sun W. Pessimistic model-based offline reinforcement learning under partial coverage[J]. arXiv preprint arXiv:2107.06226, 2021. \
> > > [5] Hornik K, Stinchcombe M, White H. Multilayer feedforward networks are universal approximators[J]. Neural networks, 1989, 2(5): 359-366.\
> > > [6] Yu, T., Thomas, G., Yu, L., Ermon, S., Zou, J.Y., Levine, S., Finn, C. and Ma, T., 2020. Mopo: Model-based offline policy optimization. Advances in Neural Information Processing Systems, 33, pp.14129-14142.\
> > > [7] Schmidt-Hieber, J. (2020). Nonparametric regression using deep neural networks with ReLU activation function. The Annals of Statistics, 48(4).\
> > > [8] Farrell, M. H., Liang, T., & Misra, S. (2021). Deep neural networks for estimation and inference. Econometrica, 89(1), 181-213.\
> > > [9] Imaizumi, M., & Fukumizu, K. (2019, April). Deep neural networks learn non-smooth functions effectively. In The 22nd international conference on artificial intelligence and statistics (pp. 869-878). PMLR.\
> > > [10] Zhou, Y., Shi, C., Li, L., & Yao, Q. (2023). Testing for the Markov property in time series via deep conditional generative learning. Journal of the Royal Statistical Society Series B: Statistical Methodology, 85(4), 1204-1222.\
> > > [11] Rashidinejad P, Zhu B, Ma C, et al. Bridging offline reinforcement learning and imitation learning: A tale of pessimism[J]. Advances in Neural Information Processing Systems, 2021, 34: 11702-11716.\
> > > [12] Yin M, Wang Y X. Towards instance-optimal offline reinforcement learning with pessimism[J]. Advances in neural information processing systems, 2021, 34: 4065-4078.\
> > > [13] Cui Q, Du S S. Provably efficient offline multi-agent reinforcement learning via strategy-wise bonus[J]. Advances in Neural Information Processing Systems, 2022, 35: 11739-11751.\
> > > [14] Wang X, Cui Q, Du S S. On gap-dependent bounds for offline reinforcement learning[J]. Advances in Neural Information Processing Systems, 2022, 35: 14865-14877.\
> > > [15] Bian, Z., Shi, C., Qi, Z., & Wang, L. (2023). Off-policy evaluation in doubly inhomogeneous environments. arXiv preprint arXiv:2306.08719.

---

### Official Review · Reviewer_GM6x · 2024-07-09

**Soundness:** 3
**Presentation:** 4
**Contribution:** 3
**Rating:** 7
**Confidence:** 3

**Summary:**

This work proposes a novel technique for performing off-policy evaluation (OPE) in the presence of unobserved confounders that have been classified as "two-way" unmeasured confounders, viz., by assuming that there exist both time-invariant and trajectory-invariant confounders, but not time-and-trajectory-invariant confounders. Roughly outlined, their approach is to first use a network architecture that takes embeddings of these different confounder types, uses a neural tensor network to first obtain estimates of transitions from a transition network and actions from an actor network, and then performs OPE in standard Monte Carlo simulation by estimating the expected cumulative reward from many simulated trajectories. Experiments compare their TWD technique's LMSE and LMAE against a host of traditional and modern techniques that span numerous means of dealing with confounding from traditional model-based methods to POMDPs to TWDIDPs. Across several simulation studies using principled data generation (like the PK-PD model for tumor growth generation), the TWD technique is shown to be the most robust and obtain lower LMSE and LMAE compared to the others compared. Ablation studies following these show how the sum of the TWD components are needed to obtain maximal accuracy.

**Strengths:**

- The review of related work is particularly rich; the authors do a tremendous job at finding recent and relevant work related to the problem space and the various viewpoints in approaching deconfounding.
- The writing is exceptionally clear, with well-stated assumptions, defined variables, and illustrations of process in each figure.
- Classifying the types of transience to which UCs belong is interesting and exploited in a nice way within the TWD technique.
- I'm a fan of the idea of performing MC simulation using the learned transition network to accomplish the OPE. I found the elaboration on line [205] to be particularly compelling.
- Compared strategies in Section 5 are comprehensive and recent.

**Weaknesses:**

- [197] One minor point that likely deserves some consideration in general: the causal structure of the system, viz., that certain contexts O should *not* be included in evidence while estimating \eta^\pi -- for example, considering the classic confounding M structure (see Cinelli, Forney, & Pearl, 2022), there are cases where an observed context must be omitted from conditioned evidence to obtain an unbiased estimate.
- [5.1] While I completely agree with the risks of ignoring UCs and that the NUC assumption is usually made for convenience rather than realism, would it not also be fair to run an experiment in an environment *without* UCs to see if there is some tax paid by this paper's technique (which seems to *assume* their presence)? It's possible that looking for confounding where none exists comes with a price that the experiments herein should advertise if present. Still, I think the authors did a good job of putting their technique through its paces within the contexts of Section 5, especially the ablation study.

**Questions:**

** Trivial Points **
- Although it has its own section explaining it, two-way unmeasured confounding appears in the introduction without even a short explanation of its meaning; it would aid the clarity if it had a brief, intuitive lead before it gets referenced in the text.
- [39] "However, these works require restrict mathematical assumptions that are hard to verify..." -- was "restrict" meant to be "strict" or "restricting"?
- [117] "...which pluggs in the estimated latent factors..." -- typo?
- [769] "A illustrative example..." -- typo
- [Eqn. 1] Why is the subscript on O_{i,1} rather than O_{i,t}?
- [236] "Assumptions 4 is concerned..." -- typo

** Substantive Questions **
- [102] "Our objective lies in evaluating the expected cumulative reward under a given target policy π, which depends solely on the observation and not on the latent confounders." The phrasing could be a bit improved here -- one read is that the expected cumulative reward depends solely on the observation and not the latent confounders, but your transition model in the previous paragraph makes it clear that this is not the case. You mean to say that the evaluation itself must be only a function of the observations since we do not have direct access to the UCs?
- [Figure 1a] What's the significance of the edges emanating from Z_{i,t}? Just to show that OWUC vs TWUC are different ways of modeling the more general unconstrained UC space?
- One part that you might want to develop a bit more: the intended application of these techniques -- is this primarily suited for artificially generated policies or those employed by human decision-makers like given in the examples?

---

> ### Author Rebuttal · Authors · 2024-08-06
>
> - __Omitting observations for avoiding biases.__\
> This is an excellent comment! We have also carefully read [1] that you mentioned and will include it as well as our discussions in our paper, shall it be accepted. This paper discusses whether an additional random variable $Z$ should be included in the regression equation for estimating the average causal effect (ACE) of a treatment $X$ on an outcome $Y$ in causal inference.
>
>   In our current problem, $\eta^{\pi}$ represents the expected cumulative reward under a target policy $\pi$. To  draw a parallel with the causal inference problem, our action $A_{i,t}$ corresponds to the treatment $X$, the expected reward $R_{i,t}$ and the next observation $O_{i,t+1}$ correspond to the outcome $Y$, and the initial observation $O_{i,1}$ corresponds to $Z$. In this case, the causal relationship among $O_{i,1}$, $A_{i,t}$, and $(R_{i,t}, O_{i,t+1})$ aligns with "Model 1" in [1], where $A_{i,t} \rightarrow (R_{i,t}, O_{i,t+1})$ and $A_{i,t} \leftarrow O_{i,1} \rightarrow R_{i,t}$. Under these circumstances, controlling for $O_{i,1}$ or including it as an independent variable in the model is beneficial because it blocks the back-door paths. Therefore, when estimating $\eta^{\pi}$, there is no need to omit the initial observation $O_{i,1}$.
> - __Settings without unmeasured confounders.__\
> This is another excellent comment. We fully agree with the importance to investigate the tax of the proposed two-way deconfounder in settings without unmeasured confounders. Indeed, our sensitivity analysis reported in Section 5.3 compares different algorithms with different degrees of unmeasured confounding, characterized by a sensitivity parameter $\Gamma$. A large $\Gamma$ indicates a stronger effect of unmeasured confounding. When $\Gamma$ decays to zero, no unmeasured confounders exist. As shown in Figure 5:
>   - When $\Gamma=0$ where there are no unmeasured confounders, the proposed two-way deconfounder (TWD) loses its superiority when compared to standard algorithms that ignore unmeasured confounders, such as MB and TWDIDP2. This demonstrates the prices of looking for confounding where none exists.
>   - When $\Gamma=0.3$ where the effects unmeasured confounding are weak, TWD achieves similar performance to MB and TWDIDP2.
>   - Finally, when $\Gamma>0.3$, the effects unmeasured confounding becomes (moderately) strong. As a result, TWD achieves the lowest LMSE and LMAE.
>
>   Inspired by Reviewer 7kZ2, we have outlined a hypothesis testing procedure to test the degree of unmeasured confounding, in order to determine the best algorithm to use. This hybrid procedure is expected to enhance the robustness of the TWD performance across diverse settings.
> - __Minor comments.__
>   - We apologize for the typos and will correct them. "restrict" was meant to be "restrictive".
>   - Shall our paper be accepted, we will add a short explanation of the proposed two-way unmeasured confounding in the introduction.
>   - In Equation (1), the conditioning set contains variables that are not affected by policies, such as the initial observation. Inclusion of $O_{i,t}$ is problematic because these observations are influenced by the behavior policy, while Equation (1) is intended to compute the expectation under the target policy.
> - __Clarification of the dependence on the latent confounders.__\
> You are absolutely correct. The immediate reward indeed depends on the latent confounders, but the target policy we wish to evaluate is irrelevant of these confounders. We plan to adopt the following changes to make this point clearer:
>   - We will rephrase this sentence to emphasize that it is the target policy rather than the immediate reward that does not depend on latent confounders.
>   - Motivated by the comments from Reviewer P4Ev, we will include visualizations of the data generating process of the offline data and that under the  to further illustrate this point; see Figure 2 in the PDF file.
> - __Clarification on Figure 1(a).__\
> Thank you for the constructive comment! The edges indicate that both OWUC and TWUC are special cases of UUC. Specifically, the edges from $\\{Z_{i,1}\\}\_{i}$ to $\\{H_{i}\\}\_{i}$ indicate that the one-way unmeasured confounders can be viewed as special cases of unconstrained unmeasured confounders that remain the same over time. Similarly, the edges from $\\{Z_{i,t}\\}\_{i,t}$ to $\\{U_{i}\\}\_{i}$ and $\\{W_{t}\\}\_{t}$ indicate that the two-way unmeasured confounders can be viewed as special cases of unconstrained unmeasured confounders that do not involve either over time or across trajectories.
> - __The intended application.__\
> Thank you for your suggestion. Should our paper be accepted, we will further elaborate on the intended application of our method, which is primarily designed for settings involving human decision-makers. In these contexts, decision-makers often rely on critical but unrecorded information when taking actions, leading to confounded datasets. For example, in healthcare or contexts similar to our case study, doctors frequently use visual observations or patient interactions to inform treatment decisions. Such unstructured data can be challenging to quantify and are often omitted  [2]. Similarly, in technology companies where behavior policies involve human interventions, the data can also become confounded [3].
> - __References.__\
> [1] Cinelli C, Forney A, Pearl J. A crash course in good and bad controls[J]. Sociological Methods & Research, 2022: 00491241221099552.\
> [2] McDonald, C. J. (1996). Medical heuristics: the silent adjudicators of clinical practice.\
> [3] Shi, C., Zhu, J., Shen, Y., Luo, S., Zhu, H., & Song, R. (2024). Off-policy confidence interval estimation with confounded markov decision process. Journal of the American Statistical Association, 119(545), 273-284.

---

> > ### Comment · Reviewer_GM6x · 2024-08-09
> >
> > Thank you for the detailed reply, to which I have only one response re: Omitting observations for avoiding biases. I appreciate you looking into the reference I provided and even citing the example of the confounding scenario with treatment $X$, confounders $Z$, and outcome $Y$. While I agree with your assessment that the observations in your work, $O_{i,1}$ are _pretreatment_ covariates (i.e., that are observed before the chosen action), I think it might be too broad to state that all such observations would match the confounding structure wherein controling for $Z$ blocks the back-door -- it is possible, as I mentioned about the M-graph example (Model 7 in Cinelli, Forney, & Pearl, 2022), to have pretreatment covariates that are correlated with both treatment and outcome but that one should _not_ control for, even if they are observed before the treatment choice.
> >
> > I also agree with your ideas for future study / amendments and think that they will strengthen this work, but am maintaining my current score (which I think is quite strong) because I cannot assess these future implementations yet, think the impact of this paper is appropriately scored, and agree with some of the theoretical qualms that some of the other reviewers have raised. Still, I think this paper introduces fresh ideas that the community would find interesting and found your replies to other reviewers to be thorough and convincing as well.
> >
> > Best of luck!

---

> > > ### Author Response · Authors · 2024-08-10
> > >
> > > Again, thank you very much for your valuable feedback and insightful comments. We are largely encouraged by your positive assessment of our paper. After carefully reviewing the paper you mentioned [1], we realize that directly conditioning on $O_{i,1}$ may have been insufficiently cautious. The issue with the M-graph that you highlighted is certainly a valid concern. We will discuss this in the paper.
> > >
> > > Meanwhile, it remains possible to employ the confounder selection algorithm to determine whether to including observations in a data-dependent manner (see e.g., Example 1 on Page 6 of [2]). However, most confounder selection algorithms only explore the bandit scenario, and further research would be required to extend this to reinforcement learning. We are truly grateful for your insightful feedback once again.
> > >
> > > [1] Cinelli C, Forney A, Pearl J. A crash course in good and bad controls[J]. Sociological Methods & Research, 2022: 00491241221099552.\
> > > [2] Guo F R, Lundborg A R, Zhao Q. Confounder selection: Objectives and approaches[J]. arXiv preprint arXiv:2208.13871, 2022.

---

### Official Review · Reviewer_7kZ2 · 2024-07-10

**Soundness:** 3
**Presentation:** 3
**Contribution:** 3
**Rating:** 7
**Confidence:** 4

**Summary:**

This paper studies the problem of confounded OPE. The authors explore a new structural assumption of the data-generating process called the two-way confounding assumption. They also propose an algorithm called the two-way deconfounder that can deal with the new setting they consider. The authors perform theoretical and empirical analyses to validate their proposed assumptions and algorithms.

**Strengths:**

* The idea of two-way confounding is interesting and well-motivated. This paper is the first to introduce the two-way confounder assumption into the field of causal RL.

* The two-way deconfounder algorithm is simple and intuitive. And it can deal with challenging causal RL problems with modern functional approximation techniques.

* The authors perform solid theoretical analysis and empirical analysis to evaluate their methodology. The theoretical results include a non-asymptotic error bound of the two-way deconfounder estimator. I checked the proof and found the result is reliable. In the empirical analysis, the authors test their method in both simulated and real-world RL scenarios. They also compare their method with various baselines.

**Weaknesses:**

* I suggest the author add more explanations about the problem setting to the intro section, making it more readable for readers outside the field of causal inference.

* The algorithm imposes additional modeling restrictions to the MDP by parameterizing $P$ as conditional Gaussians.

* (Minor) It seems that some notations may denote both a variable or a set of variables, which is a little bit confusing.

**Questions:**

* In the literature of model-based RL, the transition dynamic can be represented with some kind of generative model (for example, [1]). For the two-way confounder algorithm, is it possible to parameterize $P$ as some generative model and thus eliminate the conditional Gaussian restriction?

* Given different confounding assumptions (unconstrained/two-way/one-way confounding), how can we determine which setting fits best with the problem if we do not have full prior knowledge of the data-generating process? Do there exist any hypothesis testing procedures or simple heuristic methods? Can the authors give a short discussion?

* (Minor) What does the term "relational neural network" mean?

[1] Buesing, Lars, et al. "Learning and querying fast generative models for reinforcement learning." arXiv preprint arXiv:1802.03006 (2018).

**Limitations:**

The authors have addressed the limitations of their paper in the appendix.

---

> ### Author Rebuttal · Authors · 2024-08-06
>
> - __Enhancing clarity of the problem setting in the Introduction.__\
> Thank you for your valuable suggestion. We will include additional explanations about the problem setting in the introduction section of the final version.
> - __Clarifying confusing notations.__\
> We apologize for any confusion caused by the misused notations. We will correct these issues in the final version.
> - __Can the transition dynamics be represented using a generative model instead of a conditional Gaussian distribution?__\
> This is a good point. We did assume the transition function $\mathcal{P}$ follows a conditional Gaussian distribution in our implementation. This assumption is common employed in RL, as seen in papers such as [1] and [2]. As you mentioned, it is also possible to parameterize the transition function using a generative model, such as mixture density networks, generative adversarial networks or diffusion models. This alternative modeling approach would not affect our theoretical results, as our theory does not require the conditional Gaussianity assumption. We will add these discussions in the final paper.
> - __How can we determine the best-fitting setting for the problem without prior knowledge of the data-generating process?__\
> We greatly appreciate this excellent comment. Our paper covers four confounding assumptions, corresponding to no unmeasured confounding, one-way unmeasured confounding, two-way unmeasured confounding and unconstrained unmeasured confounding. In practice, we may consider the following sequential testing procedure to infer the unknown confounding structure and select the most suitable confounding assumption:
>
>    * **Step 1: Initial testing for the Markov property**. At the first step, we use state-of-the-art test procedures, such as [3], to determine whether the original data satisfies the Markov property. If the null hypothesis is not rejected, we do not have sufficient evidence to believe there are unmeasured confounders. Thus, we stop the procedure and conclude the data is likely does not contain unmeasured confounders. If the null hypothesis is rejected, it suggests the presence of some unmeasured confounders and we proceed to Step 2.
>    * **Step 2: Testing for one-way unmeasured confounding**. We next assume the data contains one-way confounders, estimate them from the data, include the estimators in the observations, and perform the Markov test again using the transformed data. If the null hypothesis is not rejected, we stop the procedure and conclude that the one-way unmeasured confounding assumption is likely to hold. Otherwise, we proceed to Step 3.
>    * **Step 3: Testing for two-way unmeasured confounding**: Finally, we impose the two-way confounding assumption, estimate these confounders from the data and apply the Markov test to the transformed data with estimated two-way confounders incorporated in the observations. If the null hypothesis is not rejected, we stop the procedure and conclude that the two-way unmeasured confounding assumption is likely to hold. Otherwise, we conclude that the unconstrained unmeasured confounding assumption is likely satisfied.
>
>   We will add the related discussions shall our paper be accepted.
> - __Clarification on relational neural network.__\
> The term "relational neural network" has the same meaning as "neural tensor network" [4] mentioned on line 173, known for its ability to capture the intricate interactions between pairs of entity vectors. We apologize for the confusion and will use "neural tensor network" consistently shall our paper be accepted.
> - __References.__\
> [1] Yu T, Thomas G, Yu L, et al. Mopo: Model-based offline policy optimization[J]. Advances in Neural Information Processing Systems, 2020, 33: 14129-14142.\
> [2] Janner M, Fu J, Zhang M, et al. When to trust your model: Model-based policy optimization[J]. Advances in neural information processing systems, 2019, 32.\
> [3] Shi, C., Wan, R., Song, R., Lu, W. and Leng, L., 2020, November. Does the Markov decision process fit the data: Testing for the Markov property in sequential decision making. In International Conference on Machine Learning (pp. 8807-8817). PMLR.\
> [4] Richard Socher, Danqi Chen, Christopher D Manning, and Andrew Ng. Reasoning with neural tensor networks for knowledge base completion. Advances in neural information processing systems, 26, 2013.

---

> ### Comment · Reviewer_7kZ2 · 2024-08-12
>
> I want to thank the authors for their detailed response, which addresses my major concerns. In particular, the authors present a well-rounded discussion on how to determine the best-fitting setting for the problem without knowing the data-generating process. I have also read the authors' responses to other reviews. From my viewpoint, these responses are convincing and provide interesting ideas beyond the paper. In conclusion, I think this paper is a solid work and will keep my positive rating.

---

> > ### Author Response · Authors · 2024-08-14
> >
> > We sincerely appreciate your valuable feedback, which has greatly contributed to our understanding and will guide our future work. Thank you for your time and effort in reviewing our paper!

---

### Official Review · Reviewer_wVkG · 2024-07-13

**Soundness:** 3
**Presentation:** 2
**Contribution:** 3
**Rating:** 6
**Confidence:** 4

**Summary:**

The authors study off-policy evaluation for longitudinal data with hidden confounders, which are accounted for by assuming they are either time-invariant or state-invariant but not both.

**Strengths:**

* The idea is relatively original, important, and the contribution might be significant. Presentation is somewhat clear.
 * The section studying real-world MIMIC III data shows that this method might be practically relevant.

**Weaknesses:**

* Despite the term "deconfounder" being featured prominently, the authors do not engage with much of the literature on the deconfounder algorithm initially proposed by Wang and Blei (2019). Not much is said beyond the cryptic statement, "the validity of the deconfounder algorithm relies crucially on the consistent estimation of the latent factors." This method is also inspired by two-way fixed effects regression, but since they are drawing connections to the deconfounder algorithm, the authors should discuss its fundamental assumptions in the context of its limitations and criticisms by D'Amour, Ogburn et al., and others.

 * The assumption of two-way unmeasured confounding is illuminated with concrete examples starting at line 127, but the Propositions 1-3 that follow are difficult to understand without explicit context. Looking at Appendix A, it seems that additional assumptions need to be made for Proposition 2 or Proposition 3 to be true.

Minor comments
 * Typo on line 117: "pluggs"
 * Figure 5 is too small to read.

**Questions:**

In Figure 4, why are TWD's estimated values so much more negative? And why is the variance higher?

**Limitations:**

The authors should discuss limitations in the main text rather than the appendix.

---

> ### Author Rebuttal · Authors · 2024-08-06
>
> - __Engagement with the deconfounder literature, limitations and criticisms of [1]__\
> This is an excellent comment. A detailed discussion of the criticisms of the deconfounder algorithm also helps better motivate our algorithm. We plan to include the following discussion after our cryptic statement you mentioned, to more clearly elaborate on the limitations of the deconfounder:
>
>   To ensure the latent factors can be estimated precisely, [1] imposed the **Consistency of Substitute Confounders** assumption, requiring to estimate the unmeasured confounders $Z_i$ from the causes $A_i$ with certainty. However, this assumption indeed **invalidates** the derivations of Theorems 6-8 of [1], resulting in the inconsistency of the algorithm. Specifically, as commented by [2], if the event $A=a$ provides a perfect measurement of $Z$ such that there is some function $\hat{z}(A)$ such that $\hat{z}(a)=Z$, then the overlap condition fails. As a result, the ATE cannot be consistently identified. [3] expressed similar concerns towards the algorithm. Unlike [1], our algorithm does not require such an assumption. Under the RL setting, the proposed two-way unmeasured confounding assumption effectively limits the number of unmeasured confounders to $O(N)+O(T)$, which facilitates their consistent estimation when both the number of trajectories $N$ and the number of decision points per trajectory $T$ grow to infinity, avoiding the need for the unmeasured confounders to be deterministic functions of the actions.
> - __Clarifications of Propositions 1-3__\
> Let us first clarify these propositions:
>   - Proposition 1 aims to prove that when **the true model satisfies the unconstrained unmeasured confounding (UUC) assumption**, we cannot obtain consistent estimation even if **we work with the correct model assumption (i.e., UUC)**.
>   - Proposition 2 aims to demonstrate that if **the true model satisfies two-way unmeasured confounding (TWUC)** and **we erroneously assume the model satisfies the one-way unmeasured confounding (OWUC) assumption**, then our estimators would fail to be consistent. It reveals the limitations of OUC in the presence of time-varying unmeasured confounders, thus highlighting the necessity of the proposed TWUC.
>   - Proposition 3 aims to show that when **the true model satisfies TWUC and we correctly specify two-way model**, we can obtain consistent estimation.
>
>   Meanwhile, shall our paper be accepted, we will explicitly list these underlying true model assumptions and imposed model assumptions highlighted in bold, to make these propositions clearer.
>
>   Finally, there are two additional assumptions in the proof of Proposition 2:
>   - The first assumes the oracle $\zeta$ is known;
>   - The second assumes all $W_t$s are i.i.d.
>
>   Our intention was to make the main text concise and the presentation easy to follow, so we did not introduce them in the statement of Proposition 2. The first condition is not necessary to impose, as an unknown $\zeta$ would enlarge the estimator's MSE under OWUC. Shall our paper be accepted, we will remove the first condition and explicitly state the second condition in Proposition 2. We hope the aforementioned changes clarify these propositions.
>
> - __Clarification on the experimental results of Figure 4.__
>     - __The issue of higher variance:__
>    This is another excellent comment. In response, upon checking the code of a related paper [4], we discovered that our procedure omitted a crucial step—data normalization. Given the large state dimension and significant variance among the values within each dimension in the real data, this omission likely contributed to the higher variance observed in TWD's estimated values. During this rebuttal period, we have incorporated the normalization step and re-analyzed the dataset. The updated results, illustrated in Figure 3(a) of the PDF file, demonstrate that the issue with high variance has been addressed effectively, and TWD still performs well. We will modify this figure shall our paper be accepted.
>     - __The issue of negative values:__
>     The negative values arise due to the design of the reward function. In our submitted manuscript, we set the reward to $R_{i,t}=SOFA_{i,t}-SOFA_{i,t+1}$. This choice has been previously used by [5], who also reported similar negative values. During this rebuttual, we have considered two new reward functions $R_{i,t}=-SOFA_{i,t+1}$ and $R_{i,t}=23-SOFA_{i,t+1}$, which were used in [4], and conducted additional experiments under these designs. As shown in Figure 3 of the PDF file, the estimated values of TWD vary from positive to zero to negative, confirming that the appearance of negative values stems from the choice of the reward functions. We will add these discussion shall our paper be accepted.
> - __Discussion of limitations, typos and Figure 5.__\
> We will correct the typos and move the limitations of our algorithm to the main text. Figure 5 will be enlarged shall our paper be accepted.
> - __References.__\
> [1] Wang Y, Blei D M. The blessings of multiple causes[J]. Journal of the American Statistical Association, 2019, 114(528): 1574-1596.\
> [2] D’Amour A. Comment: Reflections on the deconfounder[J]. Journal of the American Statistical Association, 2019, 114(528): 1597-1601.\
> [3] Ogburn E L, Shpitser I, Tchetgen E J T. Comment on “blessings of multiple causes”[J]. Journal of the American Statistical Association, 2019, 114(528): 1611-1615.\
> [4] Yunzhe Zhou, Zhengling Qi, Chengchun Shi, and Lexin Li. Optimizing pessimism in dynamic treatment regimes: A bayesian learning approach. In International Conference on Artificial Intelligence and Statistics, pages 6704–6721. PMLR, 2023.\
> [5] Aniruddh Raghu, Matthieu Komorowski, Imran Ahmed, Leo Celi, Peter Szolovits, and Marzyeh Ghassemi. Deep reinforcement learning for sepsis treatment. arXiv preprint arXiv:1711.09602, 2017.

---

> > ### Comment · Reviewer_wVkG · 2024-08-11
> >
> > Thank you for the detailed response. I am glad that you noticed and corrected the underlying issue with Figure 4. I am raising my score accordingly.

---

> > > ### Author Response · Authors · 2024-08-11
> > >
> > > We are sincerely grateful for your thoughtful feedback and for considering our response. We deeply appreciate the time and effort you dedicated to reviewing our work.

---

### Author Rebuttal · Authors · 2024-08-06

We thank all referees for their valuable and insightful comments. We have addressed all your comments and will incorporate them shall our paper be accepted; please refer to our detailed responses to your review. Below, we would like to briefly clarify the notation $\mathbb{E}^{\pi}$ and our data generating process, in response to the comments raised by Reviewers GM6x and P4Ev.

The notation $\mathbb{E}^{\pi}[R_{i,t}]$ represents the expectation under the **interventional distribution** of $R_{i,t}$, assuming all actions are generated solely by the target policy $\pi$, irrespective of the unmeasured confounders. In other words, whatever relationship exists between the unmeasured confounders and the actions in the offline data, that relationship is **no longer in effect** when we perform the intervention according to $\pi$. More specifically, the **interventional distribution** of $R_{i,t}$ can be described as follows:
  - The initial observation is generated according to $\rho_0$;
  - At each time $t$, the action $A_{i,t}$ is determined by the target policy $\pi(\bullet|O_{i,t})$, independent of the unmeasured confounder $Z_{i,t}$;
  - The immediate reward $R_{i,t}$ and next observation $O_{i,t+1}$ are generated according to the transition function $\mathcal{P}(\bullet|A_{i,t},O_{i,t},Z_{i,t})$.

  In this setup, the unmeasured confounders affect only the reward and next observation distributions, but not the action. This differs from the offline data generating process; see Figure 2 in the PDF file for an illustration. Alternative to $\mathbb{E}^{\pi}$, we can adopt Pearl's do-operator to further clarify the expectation, shall our paper be accepted.

---

### Decision · Program_Chairs · 2024-09-25

**Decision:**

Accept (poster)

**Comment:**

This manuscript concerns off-policy evaluation in the presence of unmeasured confounders. To this end, the authors apply causal reinforcement learning to develop a "deconfounder" to effectively learn the confounding effects and the desired policy value.

The reviewers were broadly initially impressed by the quality of this work and this sentiment was only strengthened during the author-reviewer discussion period. A vigorous discussion among the reviewers ensued following the author-reviewer discussion period. The outcome of this discussion was a consensus that there was sufficient novelty and potential interest in the problems posed by this paper, as well as the proposed solutions. There was some concern that the initial results derived here may not be the strongest possible results, but this sentiment is mitigated by opening up a promising avenue of research.